# Rewritable printing of ionic liquid nanofilm utilizing focused ion beam induced film wetting

Haohao Gu [1,7], Kaixin Meng[1,7], Ruowei Yuan[1], Siyang Xiao[2], Yuying Shan[1], Rui Zhu [3], Yajun Deng[4], Xiaojin Luo[5], Ruijie Li[5], Lei Liu [5], Xu Chen[6], Yuping Shi[5], Xiaodong Wang [6], Chuanhua Duan [2] & Hao Wang [1] ✉

Manipulating liquid flow over open solid substrate at nanoscale is important for printing, sensing, and energy devices. The predominant methods of liquid maneuvering usually involve complicated surface fabrications, while recent attempts employing external stimuli face difficulties in attaining nanoscale flow control. Here we report a largely unexplored ion beam induced film wetting (IBFW) technology for open surface nanofluidics. Local electrostatic forces, which are generated by the unique charging effect of Helium focused ion beam (HFIB), induce precursor film of ionic liquid and the disjoining pressure propels and stabilizes the nanofilm with desired patterns. The IBFW technique eliminates the complicated surface fabrication procedures to achieve nanoscale flow in a controllable and rewritable manner. By combining with electrochemical deposition, various solid materials with desired patterns can be produced.

Programmable control of fluid motion over open solid surfaces at small scales[1,2] are crucial for countless technological applications such as printing[3,4], biosensing[5,6], energy generating[7,8], air water harvesting[9], and chemical synthesis[10] etc. The predominant methods resort to either topographical or chemical gradients fabricated on solid surfaces, such as micro(nano)channels[11,12], bioinspired surface structures[13–19], chemical modifications[20,21], etc. However, such methods permanently alter the solid substrate. Once fabrication finished, the flow pattern can barely be changed. Moreover, the cost of flow control increases significantly with the complexity of surface modifications. To improve the controllability and flexibility, external-field-stimuli have been recently adopted including temperature[22,23], light[24,25], magnetic[26,27] or electric fields[4,28,29]. The current external-field-stimuli methods mainly resort to the field induced surface-tension-gradients (STGs), which works well for system size below the capillary length ($\lambda_{capillary} \sim \sqrt{\gamma/\rho g} \sim 10^{-3}$ m) since the surface energy dominates the

free energy of micrometer systems. However, the spatial resolutions of such methods are restricted at microscale and fail to achieve nanoscale flow control. For instances, the reported thermal capillary films[23–25] have thickness above 5 μm and minimum line width about 30 μm. More importantly, the finger instability is usually inevitable[30] in STG based methods and jeopardizes the patterning performances.

Actually, when the liquid film thickness approaches nanometer scale, the interfacial overlapping dominates the system free energy[31–33] instead of surface tension. Such phenomenon usually termed as disjoining pressure, arising from intermolecular forces, has been employed as a simplified mechanical description of liquid nanofilm[34–36]. Therefore, to achieve patterning flow at nanoscale, the liquid film thickness needs to reach nanometer to harness the disjoining pressure to stabilize the film pattern, and also requires the spatial resolution of external-field-stimuli to attain nanometer size. Compared with thermal[25] and pyroelectrical[26,37] fields, electron/ion

[1]Laboratory of Heat and Mass Transport at Micro-Nano Scale, College of Engineering, Peking University, Beijing 100871, PR China. [2]Department of Mechanical Engineering, Boston University, Boston 02215 MA, USA. [3]Electron Microscopy Lab, School of Physics, Peking University, Beijing 100871, PR China. [4]Future Technology School, Shenzhen Technology University, Shenzhen 518118, PR China. [5]School of Materials Science and Engineering, Peking University, Beijing 100871, PR China. [6]Research Center of Engineering Thermophysics, North China Electric Power University, Beijing 102206, PR China. [7]These authors contributed equally: Haohao Gu, Kaixin Meng. ✉e-mail: wanghpku@pku.edu.cn

beams which can be readily focused to sub-nanometer size[38,39] emerge as a promising option of external-field-stimulus. For example, 120 keV electron beam has been reported to induce the stick-slip motion of water nanodroplets[39]. However, the flow speeds were slow (around 3−20 nm/s), the travel lengths were short (around 189 nm), and more importantly, the flow directions and trajectories were uncontrollable while the physical mechanism remained ambiguous. Despite of these, their results still revealed that the interaction between electron/ion beams and liquid is a prospective candidate for liquid maneuvering and can be harnessed for open surface nanofluidics.

Here, we present an unexplored concept of open surface nanofluidics called ion beam induced film wetting (IBFW) and achieve the nanoscale patterning of ionic liquid film without resorting to any surface fabrications or special electric circuits. The Helium focused ion beam (HFIB) is generated by a state-of-the-art Helium ion microscope[38] (HIM, ORION NanoFab, Zeiss), which can provide a beam spot of diameter 0.5 nm, and a wide range of dose densities. Since the HIM only operates in an ultra-high vacuum, we select ionic liquids (ILs) as the liquid material due to its ultra-low vapor pressure. We investigate the interaction between the HFIB and IL under different dose densities and discover an unreported liquid inducing mode of HFIB. The liquid inducing mode is attributed to electrical field induced ion emission and disjoining pressure which propels and stabilizes the liquid film pattern. Based on the IBFW inducing mode, we develop a nano-printing technique of IL, with film thickness down to 20−30 nm, minimal line width about 100 nm and corner radius down to 20 nm, and compare its performances with the reported methods. Besides, ILs are also known for their unique properties such as wide electrochemical potential window, high ionic conductivity, low toxicity and thermostability. These features make ILs increasingly important as electrolytes for lithium battery[40,41] and electrodepositions of various materials ranging from metal nanoparticles[42,43], metal organic complexes[44] to conducting polymer films[45]. We further demonstrate the IBFW as a versatile tool for various application fields including gas sensing circuit, in-situ chemical reaction chip, and electrochemical deposition of solid materials with desired patterns. The simplicity and versatility of IBFW technique suggests prospect in a range of liquid manipulation applications. By combining with electrochemical procedures, such technique can not only produce patterned liquid film but also solid materials which reveals possibility in nano-transistors fabrication[46], energy devices[40] and immunosensor circuit printing[47]. We expect this technique can open a new avenue for applications in nano-printing and nano-circuit manufacturing.

## Results and discussions
### Working procedures of IBFW
The experimental system consists of a solid substrate, liquid reservoir and the HIM. A clean PECVD $SiO_2$ wafer serves as the solid substrate and forms the base of liquid reservoir. A small droplet of 1-Ethyl-3-methylimidazolium Dicyanamide ([EMIM][DCA]) IL deposited on the substrate forms the liquid reservoir. The HIM provides the non-contact external stimulus, HFIB. We systematically investigate the interaction between HFIB and the IL employing different combinations of dwell time, $\tau$, and scan spot pixel spacing, $s$, at constant beam current, $I = 1$ pA (Fig. 1a). When dose density $D = I \cdot \tau \cdot s^{-2} < 2.5$ ions $\cdot$ nm$^{-2}$, the HFIB exerts negligible influence on both solid and liquid samples which is the imaging mode of HIM. At extremely high dose density ($D > 500$ ions $\cdot$ nm$^{-2}$), the HFIB decomposes/etches the samples and Helium bubbling may occur (Supplementary Fig. 1). However, at moderate dose density range (2.5 ions $\cdot$ nm$^{-2} \le D < 200$ ions $\cdot$ nm$^{-2}$), the ILs are induced to flow into the irradiated area while the solid substrates remain intact (Supplementary Note 1) which is the liquid inducing mode of HIM. After the HFIB with appropriate parameters (Supplementary Table 1) scans outwards from the contact line (CTL) of reservoir (Fig. 1b), IL flows through local protrusions of CTL and

branches off to form hierarchical rivulets (Supplementary Fig. 2, Supplementary Movie 1). The liquid inducing mode of HIM has not been reported to the best of our knowledge, and can be harnessed to develop a nano-printing technique.

As an example of IBFW printing technology (Fig. 1c, d), we induce small amount of liquid from the reservoir to fabricate a rectangular film pattern (50 μm × 1 μm).

Step 1: Sample preparation. A $SiO_2$ wafer is placed horizontally as the substrate, Fig. 1c. No surface fabrication or physical mask is needed. A 0.1 μL droplet of IL is deposited on substrate as the liquid reservoir. The contact angle of the IL on the substrate is approximately 60° (Supplementary Fig. 3). Then the sample is transferred into the chamber of HIM.

Step 2: CTL identification and pattern design. Under the imaging mode of the HIM, the CTL of the droplet is identified. As the yellow dotted box marked in Fig. 1c, part of the CTL is chosen as the starting position of the film pattern (50 μm × 1 μm) which is drawn employing software Nano Patterning and Visualization Engine (NPVE). The NPVE rasterize-fills the pattern with scan spot arrays (Supplementary Fig. 4).

Step 3: HFIB irradiation and liquid inducing. Employing inducing mode of the HIM, the HFIB scans the designed area. The ion beam vector-scans the spot arrays point by point with serpentine scan style and specific beam parameters. Noteworthily, some other conditions other than beam parameters also influence the IBFW performances. The starting position of scan should be located close to or stuck into the CTL, and the scan direction should be outwards from the reservoir, while other configurations cannot achieve the best inducing performance (Supplementary Fig. 5 and Supplementary Movie 2, 3). The speed differences between liquid film propagation and ion beam scanning can explain the scan direction effects on IBFW inducing performance (Supplementary Movie 4). Also, the flood gun should be turned on. Since the helium ions carry positive charges, continuous irradiation would cause too much positive charges in the sample surface. The excess surface charges would alter the direction of the subsequent Helium ions and finally make the liquid film pattern tilted (Supplementary Fig. 6). When the point-by-point irradiation finishes, the thin liquid film is fabricated with designed pattern (Fig. 1d).

As shown in Fig. 1d, IBFW fabricated nanofilm reproduces not only the size of the designed pattern (50 μm × 1 μm), but also the straight edges and the 90° corners. Moreover, liquid film networks can be easily rewritten on the same solid substrate after the liquid is dissolved by acetone. As a result, the liquid film pathways can be reprogrammed in an on-demand manner.

To further demonstrate the prospect in complex patterning of IBFW, we fabricate a liquid film pattern making up the words "PKU COE" (Peking University College of Engineering) in Fig. 1e. The word patterns are all drawn employing NPVE without physical masks. Throughout the patterning process, the liquid is induced to flow along arbitrary geometrical tracks with curved lanes, crisscross junctions, and corners to form complicated liquid networks. The liquid film well reproduces the designed pattern. The width and corner radius of the liquid film barely change during the propagation regardless of the complexity of the pattern. Figure 1f presents an example of the minimal line width, 100 nm, IBFW can achieve. By assembly of separate scan patterns sequentially, the IBFW technique can also fabricate liquid film channel up to hundreds μm in length. Apart from continuous pattern, discontinuous liquid pattern can be fabricated by the introduction of damaging mode of HFIB. As shown in Supplementary Fig. 7a PKU pattern is separated from the flow channel that connect to the droplet reservoir.

### Working principles of IBFW
When HFIB irradiates at or near the CTL of droplet, the Helium ions generate special charges distribution[48] in the $SiO_2$ substrate, the positive surface charges induce the primary ion emission from the IL

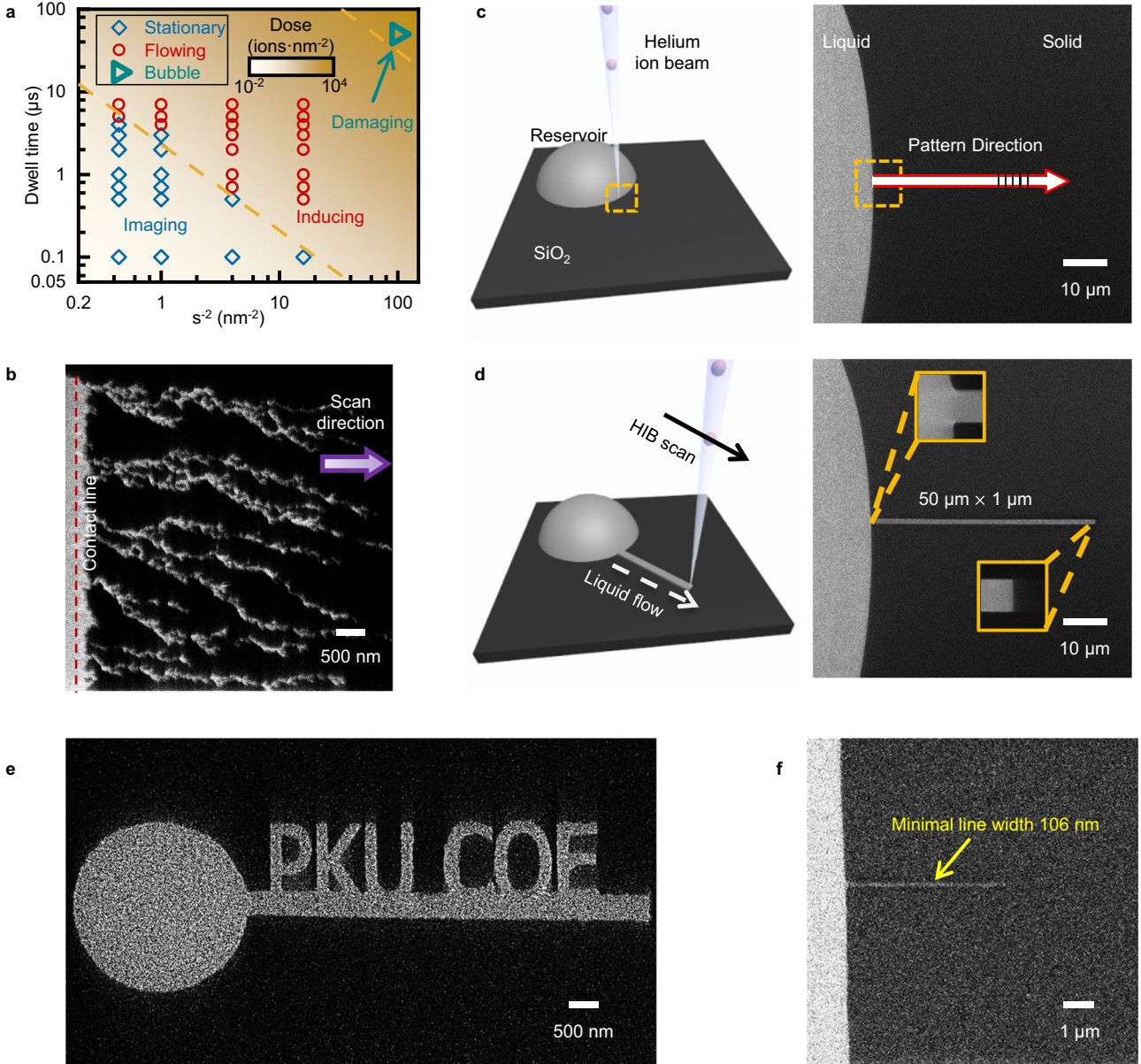

**Fig. 1 | Helium focused ion beam liquid inducing mode and working procedure of ion-beam-induced-film-wetting nano-printing. a** Phase diagram of three working modes under different HFIB dose density, manipulated by the dwell time and the scan spot density. The background color map shows the dose density ranges from $10^{-2}$–$10^4$ ions/nm². The horizontal axis corresponds to the reciprocal of the square of the scan spot spacing, $s^{-2}$, which represents the number of scan spots within 1 nm². The imaging mode exerts no influence to samples (blue squares); the inducing mode can induce directional liquid flow (red circles); and the damaging mode can etch samples and induce Helium bubbling (cyan triangle). **b** The hierarchical branched flow pattern of ionic liquid after a frame of HFIB scan. The IL will eventually fill the entire irradiated area in 30 s. **c** Contact line region of the IL reservoir and nano-patterning-and-visualization-engine pattern design. Left is a schematic diagram, and right is an HIM image. The white arrow indicates the pattern direction and area in NPVE. The dashed yellow box represents the CTL region chosen for HFIB scan. **d** The rectangular film pattern (50 µm × 1 µm) fabricated by IBFW. The insets are the details to manifest small corner radius and well reproduction of the design, the yellow box edges are 2 µm. **e** The IBFW fabricated 'PKU COE' pattern demonstrates its potential in on-demand printing of complex liquid film networks. **f** The minimal line width that IBFW can fabricate is around 106 nm.

reservoir (Fig. 2a). HFIB exhibits two distinguishable features compared with electron beam and other ion beams[48,49]. Firstly, the divergence of HFIB interacting with solid samples is much smaller than electron beam due to its larger mass[49], which lead to a more localized surface charging area and consequently a programmable control over liquid flow (Supplementary Fig. 8). Secondly, HFIB tends to penetrate sample and induce less damaging compared with Gallium beams[48]. The penetration depth exerts significant influence on IBFW's application potential in electrochemistry field. For example, 30 keV He beams with hundreds nm stopping range can easily penetrate a 10 nm Au and 5 nm Ti electrode deposited on SiO₂ wafer and induce patterned liquid flow without devastating effect to Au surface (Supplementary Fig. 9), while 30 keV Ga with stopping range less than 20 nm[48] is hard to penetrate metal films and can easily cause damage to the electrode surface. During the HFIB irradiation, He ions interact with solid atoms and excite holes-electrons in the sample, then the He ions lose kinetic energy and rest within the stopping range. Since the excited electrons in amorphous SiO₂ only survive 10 ns or less before the recombination takes place[50,51], the positive charges dominate the surface charging and account for the primary anion emission. Monte Carlo simulation

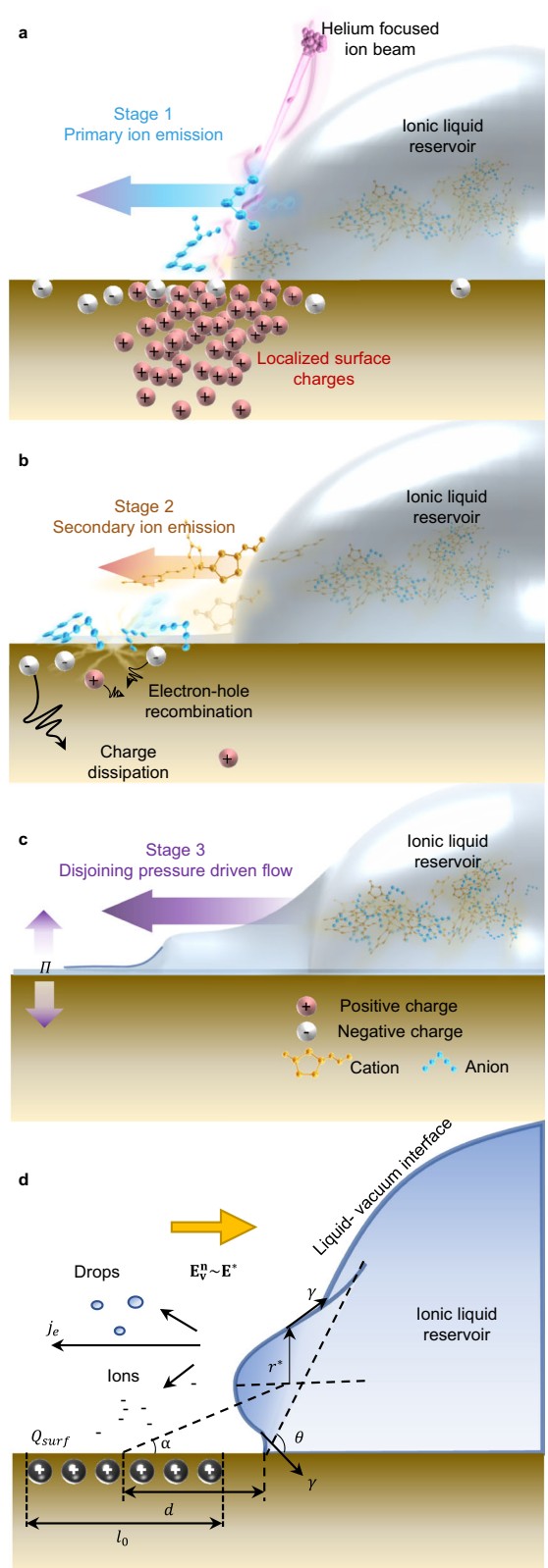

**Fig. 2 | Working principles of ion-beam-induced-film-wetting. a–c** Schematic diagrams of the IBFW working principle. **a** When helium focused ion beam irradiates the contact line, the positive surface charging induces the primary anion emission. **b** When HFIB ceases to scan, surface charging dissipates and the emitted anions induce the secondary cation emission. **c** The emitted ions from previous stages form an ultra-thin precursor film, and the consequent disjoining pressure, $\Pi$, propels and stabilizes liquid film. **d** The mechanical balance between the surface charge density induced electrostatic force and the surface tension of liquid-vacuum interface. The distance between HFIB scan spot and contact line is $d$; the surface charging uniformly distributes over a region with length scale, $l_O$; the surface-charge exerts electrostatic force and distorts the liquid-vacuum interface, balanced by the surface tension $\gamma$; $\alpha$ is the angle between substrate and the connecting line of surface-charge center and ion emission center; $r^*$ is the ion emission critical radius, and is also the vertical distance between ion emission center and substrate.

(Fig. 2b) then the emitted anions induce the secondary cation emission. Both ions meet ahead of the CTL and form an ultra-thin precursor film with thickness comparable to ion size at the irradiated area. As a matter of fact, when the electric field of surface charge is strong enough, not only separate ions but clusters or even tiny droplets contain both ions are emitted[52] to scanned area and make up the precursor films. In both cases, the ultralow thickness of precursor film gives rise to the high disjoining pressure ($10^{5\text{-}6}$ Pa) that irrigates and thickens the precursor film until be balanced by the capillary force ($10^{2\text{-}3}$ Pa) and a continuous liquid film is formed (Fig. 2c).

The surface charging process of dielectric materials ($SiO_2$ for example) under the irradiation of focus ion beams ($Ga^+$ or $He^+$) was thoroughly discussed in literatures[53–55], and the charging accumulation and dissipation is manipulated by the following factors: (1) generation of electron-hole pairs in the solid by incident ions; (2) neutralization of the incident ions by the excited free electrons; (3) sputtering of the surface atoms; (4) charging due to the secondary ion-electron emission; (5) leakage of mobile electron-hole pairs to the silicon substrate; (6) induced shallow traps by the incident ions and a consequent preferred trapping relative to the deep traps. The surface charge density (SCD) of $SiO_2$ at HFIB irradiation[54] can be expressed as a function of time (Supplementary Note 2):

$$\frac{dQ(t)}{dt} = P(1+\gamma_e) \cdot I(t) - k\frac{Q(t)}{\epsilon_r \epsilon_0} - \frac{7}{4} Y I(t) \cdot \Omega_0 \frac{Q(t)}{R_p} - \int_0^t J(t)dt. \quad (1)$$

The RHS composes of 4 terms. The first is the ion incident term which represents the electron-hole pairs accumulation induced by the ion incident and secondary emission, where $P$ is the probability factor accounts for the electron-hole recombination, $\gamma_e$ is the secondary electron emission yield of $SiO_2$, and $I(t)$ is the beam current density of HFIB. The second term is the leakage current term, where $k$ is the conductivity of the substrate, $\epsilon_r$ is the substrate relative permittivity, and $\epsilon_0$ is the vacuum permittivity constant. The third term is the sputtering yield induced charge reduction, where $Y$ is the sputtering yield acquired from SRIM simulation, $\Omega_0$ is the atomic volume which can be estimated by the average density of $SiO_2$, $R_p$ is the ions stopping range from SRIM. The last term accounts for the accumulation of emitted counterions, where $J(t)$ is the ion emission rate at current SCD, which is often described as a kinetic process in which ions evaporate from liquid-vacuum interface. The emission current density reads[56]:

$$j_e = \frac{J}{\pi r^{*2}} = \frac{k_B T}{h} \sigma \exp\left(-\frac{\Delta G - G(\mathbf{E_n^v})}{k_B T}\right), \quad (2)$$

where $j_e$ is the current emitted per unit IL-vacuum surface area, $k_B$ is Boltzmann's constant, $T$ is the liquid temperature, $h$ is Planck's constant, $\sigma$ is the local net charge density at the liquid-vacuum interface, $\Delta G$ is the Gibbs free energy barrier for an ion to be emitted,

results[48] reveal that the lateral projection length of 30 kV He ions irradiated on $SiO_2$ (also $SiO_2$ with 10 nm Au and 5 nm Ti layers) ranges from 90 nm to 110 nm (Supplementary Fig. 9a, b). The consequent positive charges (ions, holes) distribute over $10^2$ nm in lateral direction and determine the ion emission and IBFW spatial resolution.

When the HFIB ceases to irradiate, the surface charges dissipate due to the drainage current and the electron-hole recombination

$\mathbf{E_n^v}$ is the local vacuum electric field normal to the interface. $G(\mathbf{E_n^v})$ is the reduction of solvation energy barrier due to the external electric field, assumed to take the form $G(\mathbf{E_n^v}) = \sqrt{\frac{q^3 \mathbf{E_n^v}}{4\pi\epsilon_0} \frac{\epsilon_r - 1}{\epsilon_r + 1}}$ by the Schottky hump, where $q$ is the ion's charge. The solvation energy of emitted ion can be estimated by the Born model as $\Delta G = \left(\frac{27}{4}\pi\right)^{1/3} \frac{\gamma^{1/3} q^{4/3}(1-\epsilon_r)^{2/3}}{(4\pi\epsilon_0)^{2/3}}$, where $\gamma$ is the liquid–vacuum surface tension. Adopting the mechanical model proposed in the following paragraph, the electric field of SCD, $\mathbf{E_n^v}$, can help to calculate the ion emission rate and the SCD.

A schematic diagram is shown in Fig. 2d to model the mechanical balance between IL surface tension and the electrostatic force exerted by the surface charge. The IL–vacuum surface is distorted by the electric field of surface charges and forms a bumping meniscus. When the meniscus is distorted to be hemispherical, the vertical component of surface tension reaches maximum. Once the surface charge continues to increase, a significant ion emission would take place during which both ions, clusters and tiny droplets may emit from the interface[52,56,57]. The distance between the scan spot (center of surface charge) and the contact line ranges from $10^0$ to $10^2$ nm, as long as the surface charge is strong enough to induce ion emission. The critical

SCD that can induce significant ion emission depends on the distance between the starting scan spot and the reservoir CTL (more detailed deduction can be found in Supplementary Note 3):

$$Q_{surf} \cong \frac{(d + r^*)^2 + l_0(d + r^*)}{k_0 \cos^3 \alpha} \mathbf{E^*}. \tag{3}$$

Where $d$ is the distance between the scan spot and the CTL; $r^* = \frac{q^6 \gamma}{4\pi^2 \epsilon_0^3 (\Delta G)^4} \sim 10^{-8}$ m is the characteristic ion emission radius derived in literature[52]; $l_0$ is the surface charging area length scale, which represents the lateral distribution of positive surface charges[48]; $k_0$ is the Coulomb constant; $\alpha$ is the ion emission angle as depicted in Fig. 2d; and $\mathbf{E^*} \sim 10^{9 \sim 11}$ V/m is the characteristic electric field[56] for significant ion emission. At given separation distance $d$, the critical SCD can be calculated by Eq. (3). By invoking Eqs. (1) and (2), the dosage of HFIB required for the critical SCD can be calculated.

The relationship between critical HFIB dose density to induce IBFW, and the distance between starting scan spot and droplet CTL, $d$, is calculated employing Eqs. (1–3) with the results depicted in Fig. 3a by blue line. The beam current employed in calculation is 1 pA, scan

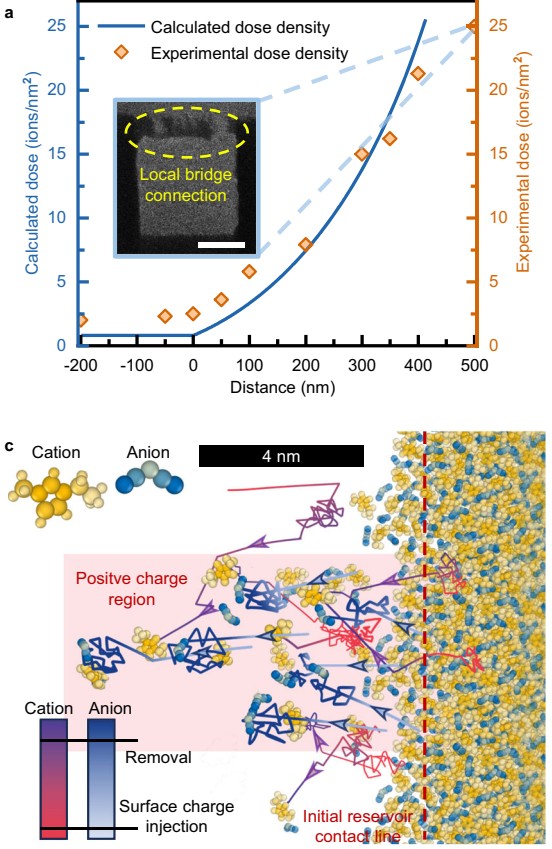

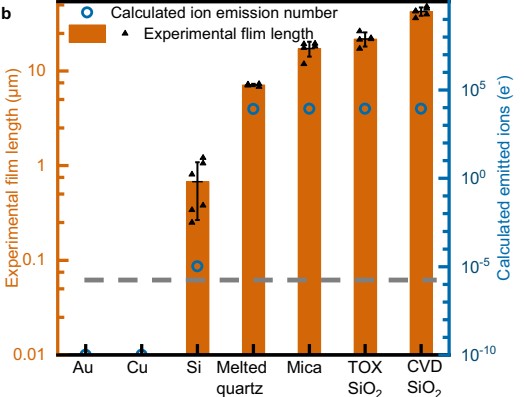

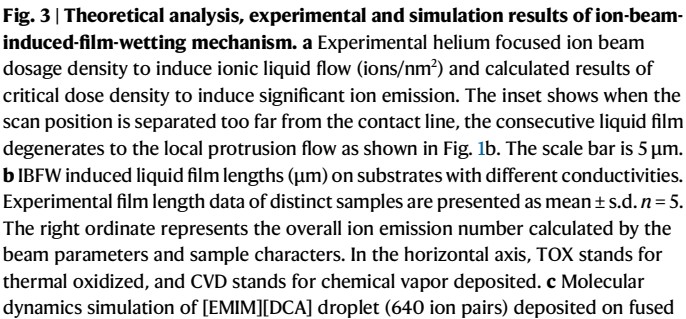

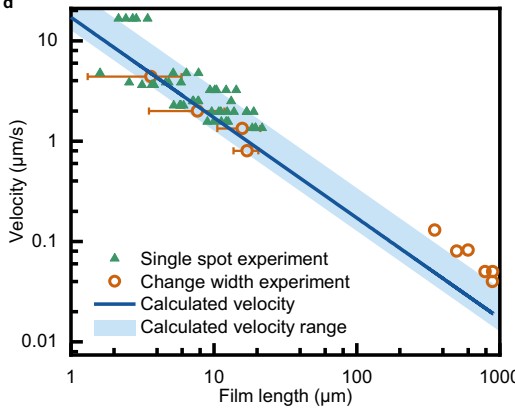

**Fig. 3 | Theoretical analysis, experimental and simulation results of ion-beam-induced-film-wetting mechanism. a** Experimental helium focused ion beam dosage density to induce ionic liquid flow (ions/nm²) and calculated results of critical dose density to induce significant ion emission. The inset shows when the scan position is separated too far from the contact line, the consecutive liquid film degenerates to the local protrusion flow as shown in Fig. 1b. The scale bar is 5 μm. **b** IBFW induced liquid film lengths (μm) on substrates with different conductivities. Experimental film length data of distinct samples are presented as mean ± s.d. $n = 5$. The right ordinate represents the overall ion emission number calculated by the beam parameters and sample characters. In the horizontal axis, TOX stands for thermal oxidized, and CVD stands for chemical vapor deposited. **c** Molecular dynamics simulation of [EMIM][DCA] droplet (640 ion pairs) deposited on fused

silica substrate going through surface charges injection and removal. The arrows indicate the most directed movements of ions: the pale blue arrows at the beginning stage represent the surface charge induced primary anion emission; the purple arrows of cations represent the emitted anions induced secondary cation emission. **d** IBFW film flow speed as a function of liquid film length. The single spot experiments are conducted with a line pattern of scan spots, at constant beam current 1 pA, dwell time 10 μs, spacing 1 nm. The change width experiments are conducted with rectangular pattern with constant length 20 μm and different widths, while keeping the beam current 1 pA, dwell time 2 μs, spacing 1 nm. Data of change width method are presented as mean ± s.d. $n = 12$. The blue shaded region is the calculated velocity range due to the range of slip lengths[58,59], the deep blue line is the calculated result employing the average slip length[58].

spacing is 1 nm, and the dwell time determines the calculated dose density. Here, we assume that the critical dose density that takes to induce significant ion emission coincides with the dose to achieve IBFW. The experiments are conducted to measure the critical HFIB dose density with 1 pA beam current and 1 nm spacing to induce continuous liquid film from reservoir with the results shown in Fig. 3a by orange rhombuses. The calculation results agree with the experimental critical dose density qualitatively, and confirm our hypothesis that the surface charging induced ion emission accounts for the IBFW.

Apart from the HFIB configuration, the substrate characteristics also influence the IBFW behavior. As shown by Eq. (1), the electrical conductivity is easy to manipulate while exerts significant effect on the liquid inducing performance. In Fig. 3b, we select 7 different solid substrates vary from well conductors to dielectrics. The orange rectangles and black triangles represent the lengths of liquid film that can be induced on different substrates under identical HFIB treatment. IL film cannot be induced on pure conducting samples such as Au and Cu, liquid extends slightly on semiconductor (Si), whereas liquid film propagates a long distance on insulated substrates *e.g.*, quartz and mica. Employing Eqs. (1–3), we can calculate the numbers of emitted ions during a single spot scan of HFIB (Supplementary Note 2, Supplementary Fig. 10), and is depicted in Fig. 3b as blue circles. As can be seen, the ion emission model we propose herein can qualitatively explain the different effects of IBFW on conducting, semi-conducting and insulating substrates. Yet the IBFW effect is also influenced by other substrate situations (roughness, for example), and the differences between various insulated substrates are failed to be captured by the simplified 2D model.

Not only the solid substrates, the ILs employed also influence the IBFW performances. Four kinds of ILs are tested on $SiO_2$ substrate with the results shown in Supplementary Fig. 11. The results show that liquid viscosity has significant influence on IBFW liquid film, by raising the flow friction which works as a counterpart of the electrostatic driving forces. The relation between wetting status and IBFW effect, however, is ambiguous, indicating that the electrostatic forces of IBFW is not quite relevant with the wetting status. It is probably due to the fact that the IBFW electrostatic forces are much greater than the wetting van der Waals forces at the contact line.

Molecular Dynamics simulation is employed to reveal the initiating stage of IBFW (Fig. 3c). The yellow spheres are atoms of cations, and blue spheres are atoms of anions as shown in the top-left legend. The bottom-left color scale of the trajectory lines represents time steps in simulation. The primary and secondary ion emissions are identified by the trajectories of anions which are represented by the white-to-blue lines and cations represented by the red-to-purple lines. The arrows indicate the most directed movements of ions. The pale blue arrows with more unified orientation at the beginning stage represent the surface charge induced primary anion emission. The purple arrows of cations represent the emitted anions induced secondary cation emission with less orientability. The red shaded regions represent the surface area where positive charges are injected and then removed. We also observe an ultra-thin precursor film in MD simulation, see Supplementary Fig. 12.

We next verify that the disjoining pressure propels and stabilizes the nanofilm. The propagation speed of IBFW liquid film decreases monotonically with the increase of film length:

$$\mathbf{U} \sim \frac{h^2 + 3bh}{3\mu} \cdot \frac{\mathbf{\Pi}(h_{\min}) - \gamma\kappa}{L}. \tag{4}$$

Where **U** is the average flow speed, $h$ is equilibrium film thickness, $b$ is the slip length of IL-SiO$_2$ interface[58–60], μ is IL viscosity, $\mathbf{\Pi}(h_{\min})$ is disjoining pressure at minimum film thickness $h_{\min}$, $L$ is film length, $γ$ is IL surface tension, and $κ$ is curvature of IL-vacuum interface at the conjunction of film and reservoir (Supplementary Table. 2,

Supplementary Note 4, Supplementary Fig. 13). We depict the calculation results in Fig. 3d. Since the boundary slip length of IL-SiO$_2$ interface depends on the combined surface conditions and ranges from 2 nm to over 16 nm[59], we employ the lower (2 nm) and upper (16 nm) limits of slip length to give an estimation on the possible range of flow speed in Fig. 3d as the blue shaded region, and the calculation result of average value 10 nm[58] is shown by the deep blue line.

The relationship between flow velocity and film length can also be measured experimentally. As shown in Supplementary Fig. 4, the HFIB scans the designed pattern row by row, so the beam speed vertical to the CTL can be calculated as:

$$\mathbf{v_{beam}} = \frac{\mathbf{s_{vertical}}}{N_{\text{row}} \cdot \tau + V_{\text{refresh}}}, \tag{5}$$

Where, $\mathbf{s_{vertical}}$ is the vertical scan spacing, $N_{\text{row}}$ is the number of scan spots in one row, $τ$ is the dwell time that HFIB stays at a single spot, $V_{\text{refresh}}$ is a small time (10 μs) that NPVE takes to reset the HFIB for next row of scan. If the vertical speed of HFIB exceeds the film velocity, the distance between the scan spot and the liquid film CTL would increases until the scan spot is too far ahead of the film and the liquid would cease to flow. The critical interaction distance with given beam parameters can be determined experimentally (Supplementary Fig. 14a). Due to the pronounced impact of dose density on the flow velocity of liquid film (Supplementary Fig. 14b), the flow velocity measurements are conducted under the same dose density by keeping the beam current $I = 1$ pA, scan spot spacing $s = 1$ nm and dwell time $τ = 2$ μs constant and only alter the width of the rectangle pattern. The pattern width controls the $N_{\text{row}}$ and consequently alters the beam vertical speed. By scanning rectangle patterns with same length but different widths outwards from the reservoir CTL, the vertical speed of beam can be changed at constant HFIB dose density. The IBFW film length decreases with the beam speed increasing (Supplementary Fig. 14c), and consequently the average flow velocity of films with different length can be measured. The change width measurement results are shown in Fig. 3d by the hollow orange circles. The results of film speed at extremely long film lengths are acquired by first fabricating a long liquid film (300 μm × 10 μm, 600 μm × 10 μm, and 900 μm × 10 μm respectively) from the reservoir, then the change width measurements are conducted at the front of the long film. Since the fabrication of extreme long film can be time consuming, these data are only measured once. The other experiments are repeated for twelve times with the average value and standard error shown in Fig. 3d.

The shortage of the change-width method is that the NPVE scan pattern assembling limits the maximum velocity the beam can move vertically. To overcome such limitation, we adopt single spot scan method. In which, a line pattern made up by a series of scan spots is used. The scan speed is altered by changing the vertical refresh time between each scan spot, while keeping beam current 1 pA, dwell time 10 μs and spacing 1 nm all constant. The dwell time is elongated to compensate the dose density reduction, since the scan area is influenced by neighboring scan spots in a rectangular pattern. All single spot measurements are repeated at least five times. The results of single spot scan are represented by the green triangles in Fig. 3d.

The consistency between the calculation and experiments suggests that disjoining pressure can explain the propagation of IBFW film. The discrepancy at extreme long film length may be due to the HFIB irradiation history. The fabrication of film with hundreds of μm length usually takes hours of HFIB irradiation. The accumulated positive charges lead to a higher surface potential and a boundary slip length that exceeds the upper limit in literature[60], which give rise to an unexpected higher film speed. Base on the film length-speed relationship curve (Fig. 3d), the fabrication limit of film length for IBFW technique can be determined to fulfill specific requirements for

fabrication efficiency. For example, the longest film we fabricated reaches 800 μm to 900 μm, with the film velocity decreases to $10^{-2}$ μm/s (the in-situ chemical reaction chip in Applications section). Besides, the equilibrium film thickness predicted by the balance between disjoining pressure[61,62] and Young-Laplace pressure agrees with the AFM measured average film thickness (Supplementary Note 5, Supplementary Fig. 15). Hence, we prove that disjoining pressure is important in the propelling and stabilizing of IBFW nanofilm.

We also exclude the potential roles played by HFIB induced surface morphological or chemical modification effects and heating effect induced Marangoni flow in Supplementary Note 1, Supplementary Figs. 9, 16, and 17. More details of MD can be found in Supplementary Note 6. Since the temperature increasing effect is less than 1 K based on our calculation. The thermal energy difference, $k_b \Delta T < 1K \cdot k_b = 8.62 \times 10^{-5}$ eV, induced by HFIB is far less than the energy barrier for ions to overcome to evaporate from the liquid phase. The evaporation effect may also be insignificant in current experiments.

## Working performances of IBFW

We employ atomic force microscope (AFM) to manifest the nanoscale flow control of IBFW. Figure 4a shows the front 7 μm of a 28 μm × 1 μm IL film with thickness around 30−40 nm (an average of 35.4 ± 1.7 nm). The film thickness remains unchanged along the flow path (Fig. 4a). The film width is 1 μm, coincides with designed pattern. The liquid-air interface is much smoother than solid substrate (RMS roughness 7.9 ± 5.7 nm). The minimal line width of IBFW film reaches 106 nm (Supplementary Fig. 18), and is limited by the surface charges spatial distribution. If the surface charges can be trapped within a narrower spatial range, the ideal line width limitation may be comparable with film thickness.

As shown in Fig. 4b, the rewritability of IBFW liquid film is tested for ten times on the same $SiO_2$ wafer. During the experiments, a 20 μm × 1 μm pattern is repeatedly fabricated at the same position each time. Then the substrate is transferred for AFM characterization (Supplementary Fig. 19) before the liquid film and droplet is rinsed for next test. Through the experiments, the film length, geometrical features and the smooth liquid-air interfaces are barely changed. Such results verify that IBFW is a rewritable open surface nanofluidic technique.

The performances of IBFW are compared with published methods[23,24] in Fig. 4c−e. The IBFW film achieves a minimal line width and thickness which are two orders of magnitude lower than those induced by thermal and optical fields (Fig. 4c). Moreover, the IBFW prevents the formation of menisci between the film and the reservoir, which are common in previous studies where the corners between the film and the reservoir CTL usually have a radius of 20−40 μm. However, we achieve sharp corners (90°) with a corner radius of 20 nm (the inset of Fig. 4d). IBFW also achieves relative high flow speed without substrate confinement. As shown in Fig. 4e, the flow speed is at least 2 μm/s at 20 μm film length at 1 pA beam current, which is one order of magnitude higher than reported methods (lower than 0.5 μm/s).

The disjoining pressure harnessed in current work explains the better performances of IBFW. We delineate a system free energy ratio scenario for a liquid film system with unit length/ width and thickness vary from 1 nm to 4 mm in Fig. 4f, and the film thickness and corner radius are compared with published results[16,18,23,24]. System free energy composes of the volumetric term (gravity, electrostatic, etc.), the surface tension term, and the disjoining pressure term. The surface tension remains fixed magnitude of $10^1$ mN/m, and its contribution to the system is almost constant with the thickness variation. When system size exceeds capillary length, $\lambda_{capillary}$, the volumetric term contributes most of the system free energy. For any system with characteristic length below capillary length, however, the influence of volumetric term (gravity etc.) can be neglected. At millimeter to

micrometer range, the surface tension dominates, and the majority of traditional microfluidics methods belong to such region, with the spatial resolution difficult to approach nanoscale. When the system size reduces to nanometer region, the interfacial overlapping[31,33] emerges. The disjoining pressure increases rapidly when film thickness approaches sub-nanometer and dominates system energy. Following the capillary length, an overlapping length can be defined as $\lambda_{overlapping} \sim \sqrt{\frac{A_{slv}}{\gamma}} \sim 10^{-9 \sim -8}$ m, where $A_{slv}$ is the Hamaker constant of solid-liquid-vacuum interfaces system. Below such size the overlapping energy contribution far exceeds any other terms, and can be termed as the characteristic length scale for nanofluidics. Neither the traditional external stimuli or surface fabrication methods can acquire nanoscale control of flow, while the IBFW approaches the limit.

## Applications of IBFW

As discussed previously, IL film pattern prepared by IBFW technology manifests three distinguishable features. First, the ultralow film thickness down to 30 nm indicates a high surface-volume ratio which is a key role in improving the gas sensing circuit sensitivity. Second, the capability of fabricating liquid film with desired pattern in a programmable and rewritable manner, which is important for in-situ chemical reaction and microfluidics chips. Third, the ILs are widely used in electrochemistry and reveal the possibility of transforming liquid film pattern into various solid materials ranges from organic to inorganic compounds.

In Fig. 5a, b and Supplementary Fig. 20, we present a room-humidity-sensing circuit utilizing IBFW technique. The water molecules are adsorbed by the IL then diffuse to and react at the electrode surfaces, generating a reaction current $I_{ds}$, at constant voltage 1 V. The source-drain currents of a IBFW nanofilm circuit and a micrometer-size droplet circuit are measured within the same chamber with the relative humidity ranging from 40% to 70%. The current in nanofilm linearly depends on the humidity, while no significant change can be observed for the micrometer droplet. The competition between the adsorption at liquid-air interface and the diffusion within liquid circuit can explain such differences. For the droplet circuit, the long distance for the water molecules to diffuse to the electrode surface restricts the reaction rate, therefore, the change of current with humidity is not obvious. Nevertheless, the thin liquid film circuit with much shorter diffusion distance and relaxation time largely accelerates the diffusion process, which is the rate-determining step of the current experiment. Moreover, the much higher surface-volume ratio benefits the adsorption of the vapor molecules and further improves the reaction current sensitivity. Both the sensitivity and response speed are greatly enhanced due to the size effect endowed by IBFW nano film. The simple device presented here can manifest the feasibility of the IBFW circuit in sensing circuit manufacturing.

To demonstrate the potential of IBFW for microfluidics chip fabrication, we design an in-situ chemical reaction micro-chip. In Fig. 5c, the schematic diagram shows a crosshair shaped micro fluid channel connects four separated droplets in four directions. Four square expansion windows are made on each part of the channel for the convenience of observation. The bottom-left inset shows the chip with four droplets on finger-tip. After the IBFW fabrications, 0.1 μL of sodium thiocyanate solution (NaSCN 0.1 M in deionized water) is injected into the top droplet, and serves as the colorimetric reagent for the detection of and in-situ reaction with different metal ions. After the injection of NaSCN, 0.1 μL of 0.1 mM $Fe^{3+}$ solution, 0.1 mM $Cu^{2+}$ solution and 0.1 mM $Co^{2+}$ are injected into the bottom, left and right droplets respectively. The microchip is rested in atmosphere for 20 min for the metal ions fully diffuse into the channels and react with SCN⁻ within different square windows with the ion names printed previously. Then the sample is transferred into vacuum chamber for 48 h to diminish the water content in the solution system, which will alter the hydration status of the metal ion complexes and improve the

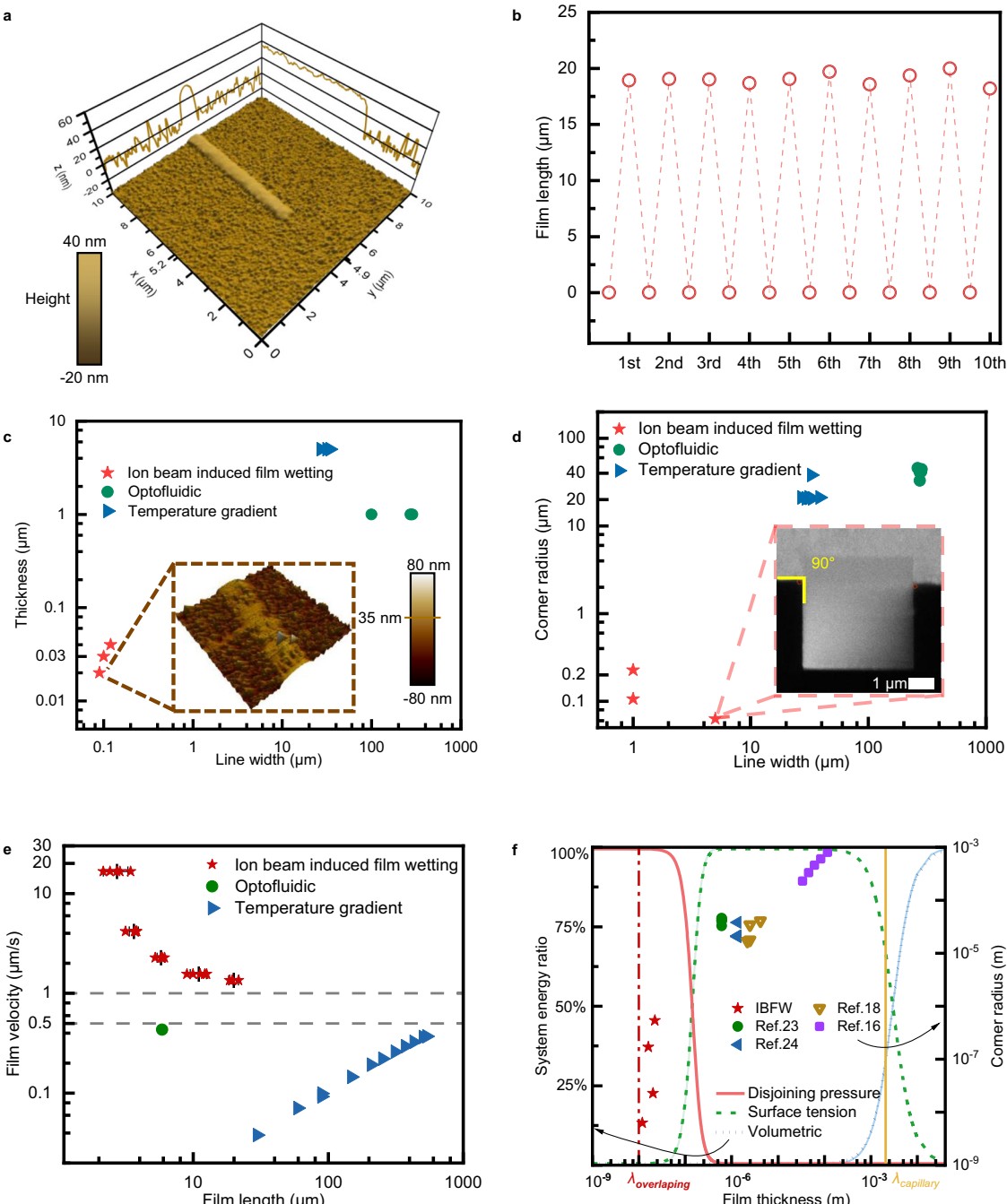

**Fig. 4 | Working performances of ion-beam-induced-film-wetting. a** Tapping mode atomic force microscope image of the front 7 μm of a 28 μm × 1 μm ionic liquid film. The height profile at y = 4.9 μm is projected onto the x-z plane, and the height profile at x = 5.2 μm is projected onto the y-z plane. **b** The rewritable test results of IBFW method on the same substrate. The same micro hole is adopted to guarantee all experiments are conducted at the same place of the same substrate. The induced film lengths of a 20 μm × 1 μm scan pattern keep constant throughout 10 times of rewritable tests. **c** The liquid film line width and thickness compared with published results[23,24]. The inset show part of the IBFW liquid film imaged by AFM. The corner radiuses between the fabricated liquid film and reservoir are

compared in (**d**) with the inset illustrates a IBFW film with corner radius down to 20 nm. **e** The flow velocities at different liquid film lengths are compared with literatures[23,24]. The data of IBFW are presented as mean ± s.d. n = 5. **f** System free energy scenario of a liquid film with different film thickness, and a comparison of different flow control strategies. The left vertical axis represents the ratio that different energy terms contribute to the total free energy. The yellow vertical line represents capillary length and the red vertical line represents overlapping length ($\lambda_{overlapping} \sim \sqrt{\frac{A_{slv}}{\gamma}}$). Different symbols represent different strategies, the IBFW pushes the limit towards nanoscale while most counterparts fall in the surface tension region.

colorimetric visibility. The red complex Fe(SCN)₃ deposits in the bottom window. The gray deposition in the left window is complex Cu(SCN)₂. And the blue deposition in the right window is complex Co(SCN)₂. As shown in Fig. 5c, the SCN⁻ participates into 3 different reactions within several hundreds of micrometers flow channel. The time series pictures are shown in Supplementary Fig. 21, and the liquid

film patterns remain unchanged during the experiments which last for over one month. The IBFW fluid channel exhibits great stabilities against vacuum/air transferring, the injection of solutions into droplet reservoir, and gravity. Such behavior demonstrates the robustness of the IBFW liquid film. More importantly, all reagents are dissolved in deionized water then injected into the IL droplets and diffuse into the

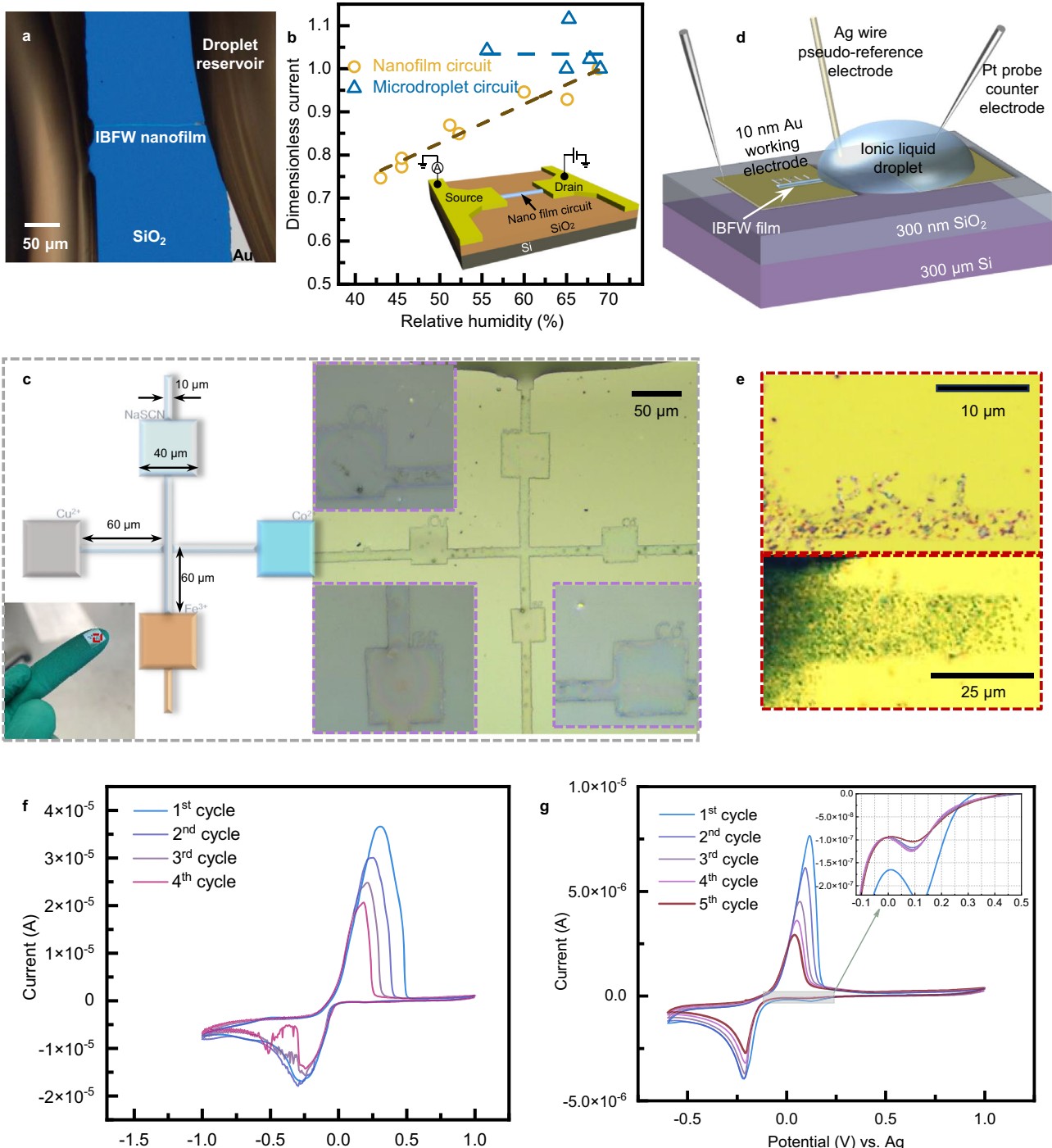

**Fig. 5 | Application examples of ion-beam-induced-film-wetting. a, b** Proof-of-concept gas sensing prototype. **a** Optical image of IBFW film that connects two droplets settled on Au electrodes. **b** Results of IBFW nano-circuit humidity sensor compared with a micrometer droplet circuit. The inset shows the schematic of the IBFW humidity sensor. The dash lines are the linear fitting results of drainage-source currents of microdroplet and nanofilm. The voltage exerted is 1 V. **c** In-situ chemical reaction chip. The left is the schematics, crosshair channels connect four droplets which are used to inject reagent water solutions. 0.1 μL of 0.1 M NaSCN, 0.1 mM $Co^{2+}$, 0.1 mM $Fe^{3+}$ and 0.1 mM $Cu^{2+}$ are injected clockwise into four droplets. The inset on the bottom-left shows the microchip on fingertip, with red circle shows the four droplets, the edge of the $SiO_2$ substrate is 10 mm. The right part is the optical image of the reaction chip that has been stored in vacuum chamber for 48 h after the injection, the purple boxes are 80 μm × 80 μm. **d** Schematics of a

three-electrode electrochemistry experiment. Analytes are dissolved in the droplet with IBFW film pattern printed onto the working electrode. 10 nm Au and 5 nm Ti serve as the working electrode, Pt probe and silver-plated probe stuck into the droplet are the counter and pseudo-reference electrodes respectively. **e** Optical images of the patterned solid particles deposited on Au electrode surface. The upper part shows Ag particles (20 μm × 3 μm rectangle with PKU letters) deposited at −0.2 V. The lower part shows blue AgTCNQ particles (50 μm × 10 μm) deposited at −0.1 V. Both are potentiostatically deposited for 180 s. Cyclic-voltammograms of (**f**) 10 mM $Ag^+$([EMIM][$NTf_2$]) and (**g**) 6 mM $Ag^+$ and 5 mM TCNQ ([EMIM][$NTf_2$]) solution. **f** The reduction peak appears at −0.295 V, with peak current decreases with cycles. **g** The first reduction peak ($Ag^+$ to Ag) appears at 0.095 V. The +300 mV shift at present of TCNQ is consistent with literature[44]. All optical images are acquired by Leica DM4B microscope.

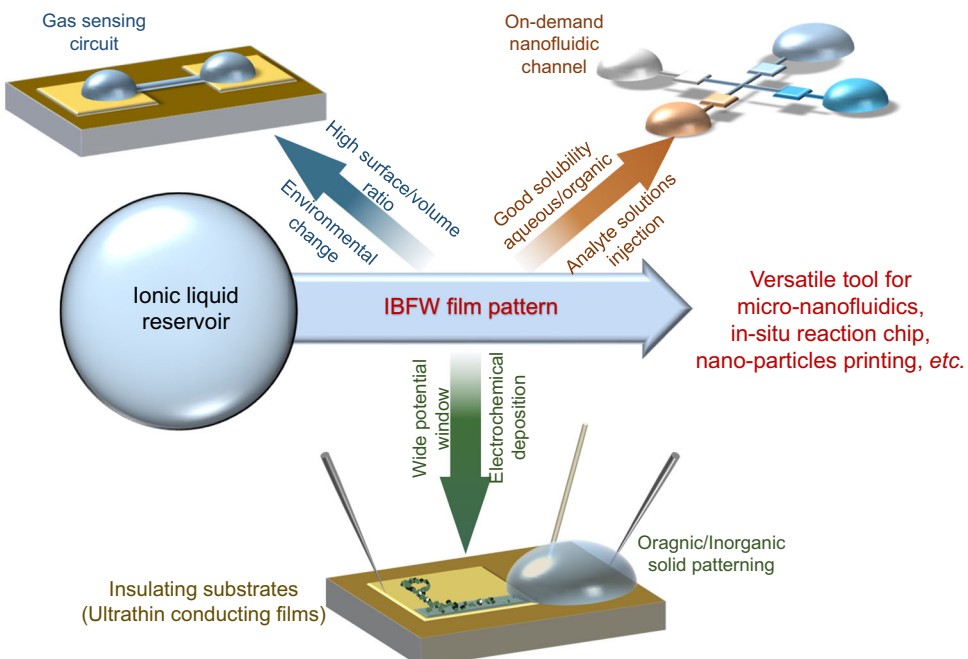

**Fig. 6 | Application potential of ion-beam-induced-film-wetting technology.** The distinct features of ionic liquid and IBFW technique manifest potential application fields. The nanometer scale film thickness makes IBFW suitable for gas sensing circuit manufacturing. The good solubility of IL and the on-demand patterning capability make IBFW suitable for nanofluidic and reaction chip fabrication. The wide potential window and stability of IL reveal possibility in transforming IBFW liquid film pattern into various solid materials.

IL flow channel. These experiments demonstrate that IBFW liquid film pattern can act as a stable flow channel for later injection of analytes dissolved in water, ethanol, and various molecular liquids due to the amphiphilicity of ILs. Such results greatly broaden the potential application fields for IBFW.

Due to their unique properties, ILs have been proved to be an important category of solvent and electrolytes. Here we demonstrate that, by further combining with electrochemical procedure, the IBFW also manifests the capability of transforming liquid film pattern into various solid materials. Figure 5d shows the schematics of a three-electrode electrochemistry experiment. A droplet of 1-ethyl-3-methylimidazolium bis(trifluoromethylsulfonyl)imide ([EMIM][NTf$_2$]) serves as the solvent of possible analytes, which are silver bis (trifluoromethylsulfonyl)imide (Ag[NTf$_2$]) or a mixture of Ag$^+$ and 7,7,8,8-tetracyanoquinodimethane (TCNQ) in our experiments. The liquid film pattern is fabricated by the IBFW technology onto the thin gold electrode and would be subsequently transformed into nanoparticles. A 10 nm Au and 5 nm Ti film deposited onto SiO$_2$ serves as the working electrode and is connected to the workstation by a Pt probe. As revealed by the MC simulation results (Supplementary Fig. 9 b), the He ions with vertical stopping-range exceeds 290 nm can easily penetrate the 15 nm metal film and deposit positive charges into the 300 nm SiO$_2$ layer underneath. Therefore, the IBFW can be achieved on an ultra-thin metal film deposited on insulating substrate, and is not contradict with the conclusion that pure conducting substrates lead to the failure of IBFW shown in Fig. 3b. The counter electrode is a Pt probe emerged in IL, and a silver-plated probe serves as the pseudo reference electrode.

In Fig. 5e, we present an example of solid particles deposited from IBFW liquid film. The upper part shows silver nanoparticles with designed pattern (a 20 μm × 3 μm channel and PKU letters) potentiostatically deposited at −0.2 V (vs. Ag) for 180 s onto the gold electrode surface. And the lower part shows blue AgTCNQ particles on gold surface make up a 50 μm × 10 μm rectangular pattern deposited at −0.1 V (vs. Ag). Noteworthily, the electrowetting phenomenon is ubiquitous in IL-electrode systems with contact angle hysteresis ranges from several to tens of degrees[63]. When the voltage applied between the IL and electrode surface is large enough to overcome contact angle hysteresis, the contact line of the IL droplet will be shifted and the IBFW patterns will be jeopardized. According to ref. 64, a negative bias voltage exceeds −1 V would induce significant contact angle decreases for [EMIM][NTf$_2$] on gold with contact line spreading forwards, which is also observed in our experiments. A negative voltage smaller than −0.5 V, on the contrary, does not influence the contact angle obviously. Therefore, the deposition voltages adopted in current work does not influence the contact line position or the IBFW liquid film.

Four cycles of cyclic voltammogram (CV) of 10 mM Ag$^+$ in IL is shown in Fig. 5f. The reduction peak of Ag$^+$ takes place at −0.29 V (vs. Ag) with the peak current decreases as the scan cycles increases. We believe that the micro liter droplet with limited analyte dissolved lead to such results. The Ag$^+$ concentration decreases quickly after each cycle of CV scan, and the electron transfer is slower for the oxidation of Ag metal. In Fig. 5g, we test the CV curves of five scan cycles of IL droplet with 6 mM Ag$^+$ and 5 mM TCNQ. Two reduction peaks can be distinguished, one at 0.095 V (vs. Ag), and the other at −0.21 V (vs. Ag). The first peak corresponds to the reduction of Ag$^+$ to Ag$^0$ (metal)[44], and the second peak is related to the formation of AgTCNQ complex (solid). Finally, we conceptually validate that IBFW technology is capable of transferring liquid film pattern into various solid materials and reveal the possibilities can be produced by combining IBFW with electrochemical procedures. The three-electrode experiment configuration, example of open-circuit-potential V-t curve and potentiostatic deposit I-t curve are shown in Supplementary Fig. 22.

As summarized in Fig. 6, the IBFW technology manifests several intriguing features that can be harnessed for a variety of application fields. The IBFW technique can fabricate patterned ILs film with 30 nm thickness, 100 nm spatial resolution and over hundreds of μm film length on insulating substrates (or coated with conducting metal films with 10$^1$ nm thickness). The surface-volume ratio endowed by the nanometer scale thickness can largely enhance the sensitivity of the IBFW film, and can be utilized in gas sensing circuit. The good solubility and biocompatibility of ILs make them suitable for the dissolve of

various analytes. The IBFW film also exhibits robustness against air environment exposure, gravity, and physical contact to droplet reservoir. Such features reveal that the IBFW film can act as stable flow channel for the analyte solutions injected to the reservoir, and can largely simplify the fabrication procedures of micro/nanofluidic chips. Last but not least, IBFW liquid film with wide potential window can be combined with electrochemical procedures and the patterned liquid film can be transformed into different solid particles. Such results demonstrate the IBFW as a versatile tool for both nanofluidics and liquid/solid materials printing.

In summary, we have demonstrated a novel strategy to control ionic liquid flow on dielectric substrates at nanoscale leveraging a novel Ion Beam induced Film Wetting mechanism. Distinct from the prevailing approaches, IBFW manipulates liquid film with minimal modification to both solid substrates and liquid. The performances of IBFW are manifested and are imparted by the utilizing of long-range intermolecular forces to stimulate and stabilize the nanofilm. IBFW can be applied to fabricate rewritable liquid circuits with intricate desired patterns for sensing and reaction applications. The combination of electrochemistry and IBFW can even transfer the liquid film pattern into solid ranges from organic to inorganic materials. Our findings will open a new avenue for versatile application fields such as on-demand manufacture of nano circuit, in-situ combinational chemistry, efficient environmental gases sensing and detection.

## Methods
### IBFW experiment
The fabrication details of the experiments used $SiO_2$ wafers are described below. An amorphous $SiO_2$ layer with a thickness of approximately 3 µm is deposited by plasma-enhanced chemical vapor deposition (PECVD) on a 500 µm thickness quartz substrate surface. Prior to the experiment, chips are cleaned with acetone, ethanol, and ultrapure water sequentially with ultrasonication. To remove any surface organic pollutants, the wafers are annealed in a tubing furnace at 320 °C with a mixture gas flow of hydrogen and argon for at least 3 h, a SUNJUNE oxygen plasma cleaner can also fulfill the purpose.

The ionic liquids used in experiments are bought from Lanzhou Greenchem ILs, LICP. Prior to the experiment, the ionic liquid is dried in a vacuum chamber at 80 °C for 24 h to remove the dissolved water. The tested ILs include 1-ethyl-3-methylimidazolium dicyanamide [EMIM][DCA] (98%), 1-ethyl-3-methylimidazolium bis(trifluoromethylsulfonyl) imide [EMIM][NTf$_2$] (98%), 1-butyl-3-methylimidazolium tetrafluoroborate [BMIM][BF$_4$] (98%), 1-ethyl-3-methylimidazolium tetrafluoroborate [EMIM][BF$_4$] (95%) purchased from Tianjin Heowns.

A drop of ionic liquid (-100 µm diameter) is deposited by the capillary tube on the top surface of the $SiO_2$ substrate acting as a liquid reservoir. Then, the sample is placed in a specimen holder and transferred into the vacuum chamber of a Zeiss helium ion microscope. The images are secondary electron images acquired by the raster scans of 30 kV He focused ion beam.

The proof-of-concept and the rewritable tests of IBFW technology employed one single PECVD $SiO_2$ wafer with preparation procedures discussed previously and a micro droplet of [EMIM][DCA] ionic liquid was deposited on the silica wafer repeatedly. After sample preparation, the solid substrate with liquid reservoir settled on the top is transferred into the HIM chamber. The CTL of droplet reservoir is identified under imaging mode, and NPVE software is employed to fabricate a 20 µm × 1 µm film pattern with the flood gun turned on to guarantee the fabricated film reproduces the pattern (the reason is discussed in Supplementary Fig. S8). After IBFW fabrication, the sample is transferred and the film pattern is characterized under the tapping mode of AFM. Once characterization is done, the solid substrate is merged into a beaker filled with acetone and placed inside a fume cupboard for 24 h. After dissolving the IL droplet and film with acetone, an ultrasonic cleaner and deionized water are employed to eliminate the residual acetone of the substrate. Such procedures were repeated for ten times in the rewritable tests.

### Wetting status and liquid viscosity effect on IBFW
The liquid viscosity influences on the IBFW performances are tested with four different ILs, [EMIM][DCA], [EMIM][BF$_4$], [BMIM][PF$_6$] and [EMIM][NTf$_2$]. Contact angles of the 4 ILs on three substrates PECVD $SiO_2$, TOX $SiO_2$, and 10 nm Au 5 nm Ti on 300 nm $SiO_2$. The contact angle measurements are conducted with a homemade platform, compose of a commercialized CCD camera (SZ-CTV, OLYMPUS), a light source and a customized metal sample platform which can adjust height and tilting angle. 1 µL of ILs droplet is deposited onto the solid surface with 1 mm capillary tube.

### Molecular dynamics simulations
The molecular dynamics simulation details can be found in Supplementary Note 6.

### Gas sensing liquid nano-circuit
The IL nanofilm circuit for gas sensing is fabricated as follow. Electrodes (Ti/Au, 5/50 nm) are deposited on doped Si wafer covered with 300 nm of $SiO_2$ by means of lithography, electron beam evaporation, and lift off procedure. Two droplets of [EMIM][DCA] IL with radius -250 µm are deposited on the electrodes acting as the liquid reservoirs. IL film connecting two reservoirs on the electrodes is obtained by the IBFW method to construct a circuit with 0.5 µm width and 80 µm length. All the transient measurements are performed at room temperature (25 °C) employing a Keithley Sourcemeter2636B, biased with a constant voltage between the source and drain electrodes. The relative humidity (RH) data are measured by a commercialized air sensor (ClearGrass air monitor, Qing Ping) with an error ±0.5%.

### In-situ chemical reaction chip
Sodium thiocyanate (NaSCN, 98.5%), cobalt(II) sulfate hydrate (100%) are all purchased from Macklin. Copper sulfate (CuSO$_4$, 98%), iron(III) chloride (FeCl$_3$, 98%) are purchased from Beijing Tongguang chemistry. The [EMIM][NTF$_2$] RTIL is chosen and four IL droplets are deposited onto a silicon dioxide chip by capillary tube to serve as the liquid reservoir for the fabrication of fluid channels employing IBFW technology. After the fabrication is finished, freshly prepared solutions are injected into the ILs droplets that have previously been deposited and connected by IBFW liquid channel. First, 0.1 µL solution of 0.1 M NaSCN in DI water is injected into the top IL droplet, and wait for 15 min for $SCN^-$ to diffuse into the entire flow channel. Then, 0.1 mM $Fe^{3+}$ solution, 0.1 mM $Cu^{2+}$ solution are injected into the bottom and left IL droplet separately. Finally, 0.1 mM $Co^{2+}$ solution is injected into the right droplet. All the injections employ a micropump. Right after the injection, the reaction chip is observed employing an optical microscope (Leica DM4 B, which is employed for all optical microscope imaging). Then the chip is stored in a vacuum chamber for 48 h, and is observed again.

The customized sample holder can adjust the tilting angle 0−180°. The reaction chip is mounted to the tilting platform with 120°, and is stored in a sample box for over one week to test the IBFW film pattern stability against gravity.

### Electrochemical depositions
Silver bis(trifluoromethylsulfonyl)imide (AgNTf$_2$ 98% EP) is purchased from Amethyst. 7,7,8,8-tetracyanoquinodimethane (TCNQ, 98% EP) is purchased from Bidepharm. [EMIM][NTf$_2$] is chosen as the solvent and electrolyte for the cyclic voltammogram and potentiostatic deposition experiments.

All the electrochemical measurements are undertaken under nitrogen atmosphere at 20−25 °C with a CHI1000C Electrochemical

Workstation (CH Instrument, Texas, USA) employing a three-electrode configuration. A homemade electrode probe platform is employed to test the micro droplet. Working electrodes (Ti/Au, 5/10 nm) are deposited on doped Si wafer covered with 300 nm $SiO_2$ by means of lithography, electron beam evaporation, and lift off procedure. A platinum probe is stuck into the IL droplet and serves as the counter electrode. An Ag plating probe serves as the pseudo-reference electrode. We do not choose a commercialized reference electrode due to their size usually far exceed the droplets used in our experiments. Prior to experiments, the working electrodes are washed with acetone, ethanol, and ultrapure water sequentially with ultrasonication, the counter electrode and pseudo reference electrode are rinsed with ethanol and DI water.

Before experiment, the IL is dried in vacuum chamber at 80 °C for 24 h to eliminate water. Then the 10 mM $AgNTf_2$ and 6 mM $Ag^+$ with 5 mM TCNQ solutions are prepared and deposited onto the silicon wafer with Au working electrodes. All these procedures are conducted in a glove box under nitrogen atmosphere. Then the silicon wafers with analyte solution droplets are transferred into the vacuum chamber of HIM for IBFW experiments. After the solution patterns are fabricated, the wafers are transferred to the electrochemical workstation under nitrogen atmosphere.

### Reporting summary
Further information on research design is available in the Nature Portfolio Reporting Summary linked to this article.

## Data availability
All data are available in the main text or the supplementary materials. Source data are provided with this paper[65–81]. Any requisition for code and technique detail should address to the corresponding author. Source data are provided with this paper.

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

## Acknowledgements

This work was financially supported by National Natural Science Foundation of China (No. 51976001, H.W.). The authors acknowledge the stimulating discussions with prof. Weidong Su and prof. Jun Xu at Peking University. The authors acknowledge the prof. Lifeng Tian at Technical Institute of Physics and Chemistry, CAS for his guidance on HIM experiments.

## Author contributions

H.G., K.M., R.Y., H.W. conceived and designed the experiments. H.W. and R.Z. supervised the study and experiments. H.G. and K.M. analyzed the experiment results and proposed the theoretical models. H.G.

conducted the MD simulation and data process coding. S.X. and C.D. fabricated and provided the silica substrates. X.L. conducted the electrical probe testing. R.L. and L.L. conducted the vacuum electrical probe testing. Y.P.S. helped with the surface annealing of the $SiO_2$ substrate. Y.D. helped with the measurements of IL contact angle. X.W. conducted the lithography fabrication of the circuit. H.G. and K.M. wrote the original draft. K.M., H.W., C.D., R.Y., Y.D., Y.Y.S. and X.C. revised the manuscript. All the authors discussed the results and provided comments.

## Competing interests

The authors declare no competing interests.
