## [Peer Review File · Nature Communications]

REVIEWER COMMENTS

Reviewer #1 (Remarks to the Author):

The present paper develops a promising technique using ion beam induced film wetting (IBFW) for the nanoscale patterning of ionic liquids film without any surface fabrications or special electric circuits. The newly-developed technique has nanoscale resolution and can be used for the open surface nanofluidics fabrication and rewritable nano print. Both the resolution and fabrication efficiency have been improved by several orders of magnitude compared to the existing technique, such as optofluidic and thermal gradient technique. Further, the authors performed systematic study on the working mechanism, the modeling and performance evaluation of the IBFM based technique, which is helpful for the fabrication parameters setup and performance improvement in the future engineering application. The present work is exciting and I suggest its publication. Furthermore, the following comments should be addressed to improve the work.

1. The key of the present work is to overcome the pinning of contact line by the electrostatic forces, which are generated by the focused ion beam. Although the authors studied the IBFW induced liquid film lengths on substrates with different conductivities, however, the effects of wetting status or surface energy of different substrates on the performances should be discussed.
2. The present work reported that the flow velocity decreases with the film length, and give a model of these two parameters. Based on the analysis, what is the fabrication limit of film length for the present technique? The authors should give necessary discussion.
3. Based on the analysis of working mechanism of the present work, the electrostatic force is important for the film generation. However, the surface charge on the substrate will also generate electric double layer (EDL) in the IL. At the same time, the thickness of film is only tens of nanometer, the EDLs on the solid-liquid interface and liquid-gas interface will overlap and generate obvious electrokinetic effect when the IF flows. What is the possible effect of EDL on the flow velocity and film generation?
4. Based on the previous MD simulation, the molecular slip happens on even smooth surface. Did the authors measure the slip on their sample pairs using MD? Because the precursor film is assumed to be 0.6 nm, the molecular slip can generate significant effect on the IL motion. This may explain the the discrepancy of flow velocity between the experiments and theory.
5. What are effects of temperature gradient and evaporation on the present technique?

Reviewer #2 (Remarks to the Author):

Referee report on the manuscript NCOMMS-23-30881 "Rewritable nano print of ionic liquids utilizing focused ion beam induced film wetting" by Gu et al.

This manuscript describes experiments demonstrating a novel way to print ionic liquids on insulating substrates using a He-ion beam writing. To my knowledge results are novel, their importance to the field of controlling liquid flows are hard for me to judge. The explanation of the working principle has several apparent deficiencies.

There are several remarks I would like to make:

-The relevance for applications was not entirely clear to me. Several references were cited in the beginning, listing a broad range of applications for “programmable control of fluid motion”. But as I understood, the present technique works for ionic liquids only. Thus, I was left wondering what the applications are specific to ionic liquids. That needs to be explained much better.

-The fabrication has to be performed in an ultra-high vacuum environment. It was not very clearly stated or studied, what happens to the samples after they are taken out of the HIM and exposed to air, tilting of the chip (gravity pulling in another direction), static charges, mechanical contact or other environmental variables. Common sense picture of liquids gives the impression that the “circuit” is not very stable. If it is, data needs to be provided to convince the reader otherwise.

-Some data was provided with different substrates. What about varying the surface treatment, or cleaning? How sensitive is the success of the writing to the cleanliness of the substrate? You say you anneal the substrates at 320 C in hydrogen flow. Is that to remove organic contaminants? What if you skip the annealing step? Or use oxygen plasma cleaning instead?

- How do you know that the ions of the ionic liquid stay intact (are not broken down) under the He ion irradiation? Typically, organic compounds (such as resists for example) can be strongly modified (bonds broken etc) with high energy ion beams.

-What sets the “minimal line width” scale of 100 nm? Shouldn't it in principle be much narrower, as the He ion beam interaction volume at the surface is very small (Fig S7) (hence its high imaging resolution < nm).

-Related to that, I do not understand the suggested mechanism. In Fig. 2A the local surface charges, induced by the beam current, are located directly under the IL droplet. Yet, in the unnumbered Figure in Note S3 (Electric field driven ion emission calculation), the surface charge is located some distance away from the IL reservoir. Something doesn't match.

-Eq. (1) cites for example Ref. 46, where some of the terms were introduced. However, in that paper, surface charge density Q was specifically described as “areal hole density” i.e. describing a positive charge, whereas you seem to assume a negative charge. How can you apply the same equations to these two opposing cases? Why would in your case the negative charge be dominant, (which the authors in Ref 46 assume can be neutralized more easily)?

-In addition, you should describe already in the main text what each term in Eq (1) describes, and define every variable of the equations of the main text (including Eq (2)).

-Shouldn't the electrons of the flood gun be somehow part of the modelling of Eq. (1) ?

To understand Eq (2), the figure from Supplementary Note 3 is required in the main text.

How did you measure the experimental surface charge density of Fig. 2 D ? Could you experimentally determine the sign of the charge? If you used Eq (1), how is that an “experimental” result?

-“The qualitative consistency ... verifies the mechanism we propose.” Big statement, considering that you refer to Fig. 2E where the agreement is NOT good. (i.e. modelling doesn't predict at all differences between different insulating substrates)

“The relationship between flow velocity and film length can also be measured experimentally”. I don't understand how you can measure the flow velocity in your experimental setting.

AFM measurements “The liquid-vacuum interface...” Is the AFM measurement done in vacuum instead of air?

“The current in nanofilm linearly depends on the humidity”. What is the mechanism?? Anyways, it seems the current is not only a function of humidity as it depends directly on time as well based on Fig. S18. How could you disentangle the effect of time from the effect of humidity in a real sensor as some voltage bias history dependence seems to exist (referring to Fig. S18F)

Fig. 3F use real current units instead of dimensionless and tell what the bias voltage used was. BTW Fig. S18 C has wrong units (A vs nA). Fig. S18G has no caption, what is it?

In conclusion, major revisions are required, in my opinion.

The impact is also not clear, so at the moment I cannot recommend publication in Nat Comm.

Reviewer #3 (Remarks to the Author):

The authors developed a new method to perform rewritable surface printing of ionic liquids. The method relies on the disjoining pressure to induce surface films of nanometer thickness, and utilizes helium ion beams to initiate the flow of liquids and pattern creation. The manuscript comprehensively characterizes and explains the underlying mechanism of this printing process. However, as an application-driven work, the prospect of practical real-world applications is a bit weak. Although the authors demonstrated a sensing device application (Fig. 3F), it only utilizes the nanoscale thickness of the ionic liquid film, and the surface patterning is not very relevant. Specific concerns include:

1. This method utilizes ionic liquids, and seems hard to extend to other liquid systems due to the high-vacuum requirements of the helium ion microscope. Ionic liquid as of now are still quite expensive, limiting the range of applications. Are there ways to improve or revise the method so that other liquid or even solid structures can also be printed?
2. The surface-wetting driven printing process is limited to patterns that are continuously connected. For printing applications, most of the time disconnected patterns are needed. Are there ways to create such disconnected patterns (in the nano/micron scale)?

If either or both of the above concerns can be successfully addressed, I suggest that this manuscript may be reconsidered. Otherwise, it may be more suitable to a more specialized journal.

Responses to Reviewer #1:

The present paper develops a promising technique using ion beam induced film wetting (IBFW) for the nanoscale patterning of ionic liquids film without any surface fabrications or special electric circuits. The newly-developed technique has nanoscale resolution and can be used for the open surface nanofluidics fabrication and rewritable nano print. Both the resolution and fabrication efficiency have been improved by several orders of magnitude compared to the existing technique, such as optofluidic and thermal gradient technique. Further, the authors performed systematic study on the working mechanism, the modeling and performance evaluation of the IBFM based technique, which is helpful for the fabrication parameters setup and performance improvement in the future engineering application. The present work is exciting and I suggest its publication. Furthermore, the following comments should be addressed to improve the work.

Response: We thank the reviewer for the positive comments on our manuscript.

Comment #1: *The key of the present work is to overcome the pinning of contact line by the electrostatic forces, which are generated by the focused ion beam. Although the authors studied the IBFW induced liquid film lengths on substrates with different conductivities, however, the effects of wetting status or surface energy of different substrates on the performances should be discussed.*

Response: We thank the reviewer for the inspiring suggestion. We have tested the wetting status of several liquid-solid combinations and include them in the Supplementary Table 4, but interestingly we find the relation between wetting status and Ion Beam induced Film Wetting (IBFW) effect is quite ambiguous, indicating that the electrostatic forces of IBFW is not quite relevant with the wetting status, which is possible since the IBFW electrostatic forces are much greater than the wetting van der Waals forces at the contact line. No significant differences were observed between same liquids on different insulating substrates (TOX/PECVD SiO₂, 15 nm metal on SiO₂). While different liquids with different viscosities showed quite different performances on the same SiO₂ substrate.

On the other hand, the liquid viscosity has significant influence, as included in Supplementary Fig. 11. The viscosity raises flow friction which works as a counterpart of the electrostatic driving forces. ILs with lower viscosity can be easily induced to flow on all 3 substrates and form liquid film with hundred μm length, such as [EMIM][DCA] and [EMIM][NTf₂]. The results in Supplementary Fig. 11 only shows the film length fabricated by one time scan of a $50\ \mu\text{m} \times 1\ \mu\text{m}$ pattern. For [EMIM]BF₄, the viscosity increases slightly and can only be induced to flow for several μm , and difficult to form desired patterns. When the viscosity increases dramatically, for example [BMIM]PF₆, helium focused ion beam (HFIB) cannot induce liquid flow on any solid surfaces.

“Not only the solid substrates, the ILs employed also influence the IBFW performances. Four kinds of ILs are tested on SiO₂ substrate with the results shown in Supplementary Fig. 11. The results show that liquid viscosity has significant influence on IBFW liquid film, by raising the

flow friction which works as a counterpart of the electrostatic driven forces. The relation between wetting status and IBFW effect, however, is ambiguous, indicating that the electrostatic forces of IBFW is not quite relevant with the wetting status. It is probably due to the fact that the IBFW electrostatic forces are much greater than the wetting van der Waals forces at the contact line.”

(Page 6-7, Main text)

Supplementary Table 3.

Parameters of 4 RTILs employed (20 °C)

Name	MW (g/mol)	Surface Tension (mN/m)	Density (g/cm³)	Viscosity (cP)
[EMIM][DCA]	177.21	47.3	1.11	21
[EMIM]BF ₄	197.97	49	1.294	45
[BMIM]PF ₆	284.18	38	1.37	284.18
[EMIM][NTf ₂]	391.31	36	1.53	32

Supplementary Table 4.

Wetting status of different RTILs on three solid surfaces (contact angle in degree, 20 °C)

Name	PECVD	TOX	10 nm Au
	SiO ₂	SiO ₂	on SiO ₂
[EMIM][DCA]	71.04 ± 6.4	40.1 ± 3.8	47.0 ± 4.5
[EMIM]BF ₄	58.9 ± 3.4	35.7 ± 4.8	55.1 ± 4.1
[BMIM]PF ₆	82.6 ± 6.3	62.9 ± 2.1	68.2 ± 8.9
[EMIM][NTf ₂]	59.2 ± 5.7	36.2 ± 3.9	37.8 ± 4.1

(Page S41-42, Supplementary)

Supplementary Figure 11: the IBFW results of different ILs and the relationship with Oh number.

(a) Film lengths of different liquids on SiO₂ substrate can be induced with a same pattern. Scan parameters are 1 pA, 2.5 nm spacing, 3 μs dwell time, 50 × 1 μm pattern. (b) The flow

velocity of different ILS. (c) The relationship between Ohnsorge number $Oh = \frac{\mu}{\sqrt{\rho d \gamma}}$ and flow velocity, the fluid data can be found in Supplementary Table 3.

(Page S28, Supplementary)

Comment #2: *The present work reported that the flow velocity decreases with the film length, and give a model of these two parameters. Based on the analysis, what is the fabrication limit of film length for the present technique? The authors should give necessary discussion.*

Response: We thank the reviewer for this important suggestion. The flow speed of IBFW film decreases with the film length, but as far as we know the flow speed does not decrease to exact zero. Therefore, the fabrication limit of film length for IBFW depends on the user's endurance for the fabrication speed of liquid film. Base on the liquid film length-speed relation curve (Fig. 2h), the film length upper limit can be determined to fulfill specific requirements for fabrication efficiency. For example, the longest film we fabricated reaches 800 μm to 900 μm , with the film velocity decreases to 10^{-2} $\mu\text{m/s}$ (the in-situ chemical reaction chip in Fig. 4c and Supplementary Fig. 20). We included an explanation in the main text as an instruction for IBFW fabrication. The optical figure of in-situ reaction chip in Supplementary Fig. 20d is a good example of IBFW liquid channel with upper limit length.

“The inset of Fig. 1e presents an example of minimal line width, 100 nm, IBFW can achieve. **By assembly of separate scan patterns sequentially, the IBFW technique can also fabricate liquid film channel up to hundreds μm in length.**”

(Page 4, Main text)

“The consistency between the calculation and experiments suggests that disjoining pressure can explain the propagation of IBFW film. The discrepancy at extreme long film length may be due to the HFIB irradiation history. The fabrication of film with hundreds μm length usually takes hours of HFIB irradiation. The accumulated positive charges lead to a higher surface potential and a boundary slip length that exceeds the upper limit in literature, which give rise to an unexpected higher film speed. **Base on the film length-speed relationship curve (Fig. 2h), the fabrication limit of film length for IBFW technique can be determined to fulfill specific requirements for fabrication efficiency. For example, the longest film we fabricated reaches 800 μm to 900 μm , with the film velocity decreases to 10^{-2} $\mu\text{m/s}$ (the in-situ chemical reaction chip in Fig. 4c and Supplementary Fig. 20).**”

(Page 8, Main text)

Supplementary Figure 20: Time series of in-situ chemical reaction chip.

(a) HIM image of channels right after the IBFW fabrication. (b) The fluid channel first transferred to the optical microscope; the colorless transparent fluid channel is a bit hard to be distinguished from the transparent silica substrate. (c) After the injection of all solutions and reacts in atmosphere for 5 min. (d) **The whole picture of the reaction chip before transferred to vacuum chamber.** (e) 48h after metal ions injection and storing in vacuum. After stored in vacuum chamber, the water has been eliminated, and the color of the different complexes become obvious. The sample goes through several times of transferring between vacuum chamber and air environment and the injection of analyte solutions into droplet reservoirs, while the liquid film pattern remains barely changed through the experiments. Such results demonstrate the stability of IBFW liquid film channel to the exposure of air environment and to non-direct physical contact for solution injections. (f) The reaction chip one month after injection (one week for gravity stability test). The chip is mounted on a customized sample holder which can adjust the tilting angle from 0-180° to test the IBFW film stability against gravity. The inset shows the mounted sample with tilting angle 120° and is stored for one week. (Page S37, Supplementary)

Comment #3: *Based on the analysis of working mechanism of the present work, the electrostatic force is important for the film generation. However, the surface charge on the substrate will also generate electric double layer (EDL) in the IL. At the same time, the thickness of film is only tens of nanometer, the EDLs on the solid-liquid interface and liquid-gas interface will overlap and generate obvious electrokinetic effect when the IF flows. What is the possible effect of EDL on the flow velocity and film generation?*

Response: We thank the reviewer for this very important comment. The EDL in IL and the

electrokinetic phenomena exert significant influence on the hydrodynamics behavior of ionic liquids, and a positive surface charge would enhance the flow velocity of ionic liquids with similar cation/anion sizes ([EMIM][NTf₂], [EMIM][DCA] in our experiments) by increasing the boundary slip length. The EDL determines the structure of the IL-solid interface, and adjusts to different surface potential (main text Ref. [58-60]). The consequent boundary slip length can vary from several nm to over 16 nm. A larger surface potential creates an absorbed ion lubrication layer which can further reduce the IL-solid interface friction, in our case this may further enhance the flow velocity. The IL-vacuum interface influence on the structure of IL-solid is negligible due to its involatile nature, we neglect its influence on the hydrodynamics. Apart from surface potential, the surface roughness may also enhance the slip length. The slip length data from literature (main text Ref. [58, 59]) of imidazolium IL-SiO₂ interface is adopted to modify our flow model. Since slip length can be easily tuned by surface potential, we employ the lower (2 nm) and upper (16 nm) limits of slip length (Ref. [59]) to give an estimation on the possible range of flow speed in Fig. 2h, and the average value 10 nm (Ref. [58]) is shown by the deep blue line. The modified flow model can better explain the high velocity range observed in IBFW experiments.

The EDL effect on the ion emission from liquid-vacuum interface may be not important. The MD simulation results indicate that the characteristic electric field (10⁹ V/m for separate ion emission, 10¹⁰⁻¹¹ V/m for ion clusters emission) that lead to significant ion emission agrees with literature results (Ref. [56]) where the ion emission directly from an IL-vacuum interface far from the solid surface.

“We next verify that the disjoining pressure propels and stabilizes the nanofilm. The propagation speed of IBFW liquid film decreases monotonically with the increase of film length:

$$U \sim \frac{h^2 + 3bh}{3\mu} \cdot \frac{\Pi(h_{min}) - \gamma\kappa}{L} \quad (4)$$

Where U is the average flow speed, h is equilibrium film thickness, b is the slip length of IL-SiO₂ interface⁵⁸⁻⁶⁰, μ is IL viscosity, $\Pi(h_{min})$ is disjoining pressure at minimum film thickness h_{min} , L is film length, γ is IL surface tension, and κ is curvature of IL-vacuum interface at the conjunction of film and reservoir (Supplementary Table. 2, Supplementary Note 4, Supplementary Fig. 13). We depict the calculation results in Fig. 2h. **Since the boundary slip length of IL-SiO₂ interface depends on the combined surface conditions and ranges from 2 nm to over 16 nm, we employ the lower (2 nm) and upper (16 nm) limits of slip length to give an estimation on the possible range of flow speed in Fig. 2h as the blue shaded region, and the results of average value 10 nm⁵⁸ is shown by the deep blue line.**”

(Page 7, Main text)

“We adopt the average slip length data $b \approx 10 \text{ nm}$ of ionic-liquid-SiO₂ interface from literature^{1,2}, and zero lateral shear stress is taken at the liquid vacuum interface. **According to the literatures, the boundary slip length of IL-solid interface depends on the specific conditions³**, such as, the liquid solid combination, the surface roughness, **the surface potential**. In most cases, the slip length increases with the roughness. The boundary slip length at different surface potential has been reported to vary due to the structure of EDL adjust to different surface

potential, ILs with higher conductivity can be influenced more easily³. A larger surface potential has been reported to have an absorbed ion lubrication layer which can further reduce the IL-solid interface friction⁴, in our case this may further enhance the flow velocity. For a better understanding of the hydrodynamics behavior of IBFW flow, both AFM measurements and MD simulation can be conducted.”

Modeling of flow velocity

The simplified 2-dimensional form of Navier-Stokes equation (4.1) is adopted and together with the zero-shear-stress condition (4.2) and the boundary slip condition (4.3) give rise to the flow velocity as a function of vertical coordination z (4.4).

$$\mu \frac{d^2 u}{dz^2} = \frac{dp}{dx} \quad (4.1)$$

$$\frac{du}{dz} \Big|_{z=h} = 0, \quad (4.2)$$

$$u(0) = b \cdot \frac{du}{dz} \Big|_{z=0}, \quad (4.3)$$

$$u(z) = \frac{1}{2\mu} \frac{dp}{dx} z^2 - \frac{h}{\mu} \frac{dp}{dx} z - \frac{bh}{\mu} \frac{dp}{dx}. \quad (4.4)$$

The flow rate can be calculated as

$$Q = d \cdot \int_0^h u(z) dz \sim - \frac{d \cdot h^2 (h+3b)}{3\mu} \frac{dp}{dx}. \quad (4.5)$$

With the average flow speed is expressed as

$$u_{ave} = Q/S \sim - \frac{h^2 + 3bh}{3\mu} \frac{dp}{dx}. \quad (4.6)$$

According to the IBFW hypothesis and the MD results, the precursor film ahead of the propagation bulk film has a thickness comparable with the size of the ion pairs of the IL, in our case the [EMIM][DCA] pair namely. An appropriate estimation for h_{min} is taken as 6×10^{-10} m, and the excess disjoining pressure of the precursor film serves as the driving force with the interface curvature induced capillary pressure serving as the resisting force, which gives a rough estimation for the pressure gradient ($\Pi(h) \ll \Pi(h_{min})$ and is omitted)

$$\frac{dp}{dx} \sim \frac{\Delta p}{\Delta x} = \frac{\Pi(h_{min}) - \gamma\kappa}{L} = \frac{\left[- \left(\frac{A_{sl} f - A_{ll} f'}{6\pi h_{min}^3} - s_p \exp\left(\frac{d_{min} - h_{min}}{l}\right) - 8c_{IL} \cdot h_{min}^{-7} \right) - \gamma\kappa \right]}{L} \quad (4.7)$$

Where s_p is the polar component of the spreading coefficient, d_{min} is the atomic cut-off distance, l is the correlation length, c_{IL} is the strength Born repulsion. The result from Note S5 is employed.

A rudimental estimation for the average flow speed *i.e.* the mesoscopic propagation speed of IBFW liquid film can be expressed as a function of the length of the flow pattern, the flow speed decreases monotonically with the increase of flow length:

$$U \sim u_{ave} = \frac{h^2 + 3bh}{3\mu} \cdot \frac{\Pi(h_{min}) - \gamma\kappa}{L}. \quad (4.8)''$$

(Page S10-11, Supplementary)

“**Figure 2: (h)** IBFW film flow speed as a function of liquid film length. The single spot experiments are conducted with a line pattern of scan spots, at constant beam current 1 pA, dwell time 10 μs , spacing 1 nm. The change width experiments are conducted with rectangular pattern of scan spots with constant length 20 μm and different width, while keeping the beam current beam current 1 pA, dwell time 2 μs , spacing 1 nm constant to keep the dose density unchanged. The blue shaded region is the range between the calculate velocity lower and upper limit due to the range of slip lengths^{58,59}, the deep blue line is the calculation result employing average slip length⁵⁸.”

(Page 22, Main text)

Comment #4: Based on the previous MD simulation, the molecular slip happens on even smooth surface. Did the authors measure the slip on their sample pairs using MD? Because the precursor film is assumed to be 0.6 nm, the molecular slip can generate significant effect on the IL motion. This may explain the the discrepancy of flow velocity between the experiments and theory.

Response: We sincerely thank the reviewer for this important insight. We have accordingly modified the flow model to slip boundary, and adopted the experimental slip length range from literatures (Ref. [58-60]). The boundary slip lengths of ILs are typically around several to tens of nanometers as reported in literatures (Ref. [58, 59]). The exact value of slip length depends strongly on the combined surface conditions including: surface roughness, surface potential and specific surface treatments. For our film system with thickness down to nanometer scale, the slip length greatly improves the calculated flow velocity at constant driving force, and agrees with our experiment results better (main text Fig. 2h). The formation of the 0.6 nm precursor film is related to the significant ion emission processes in current model, and we did not include it into the flow modeling. The discrepancy at long film length in Fig. 2h may be

due to the HFIB irradiation history. The fabrication of film with hundreds of μm length usually takes hours of HFIB irradiation, and the accumulated positive charges lead to a higher surface potential and a boundary slip length that exceeds the upper limit in literature, which give rise to an unexpected higher film speed.

We also added an illustration of the two different flow velocity measurement methods adopted in our IBFW experiment according to other Reviewer's comments, with more data measured. Which can be found in main text and Fig.2h. To help illustrate our velocity measurement, we added Supplementary Fig. 14. The modification of the liquid flow model has been quoted in our response to Comment #3 and will not be quoted repeatedly.

“We next verify that the disjoining pressure propels and stabilizes the nanofilm. The propagation speed of IBFW liquid film decreases monotonically with the increase of film length:

$$U \sim \frac{h^2 + 3bh}{3\mu} \cdot \frac{\Pi(h_{min}) - \gamma\kappa}{L}. \quad (4)$$

Where U is the average flow speed, h is equilibrium film thickness, b is the slip length of IL-SiO₂ interface^{58,59}, μ is IL viscosity, $\Pi(h_{min})$ is disjoining pressure at minimum film thickness h_{min} , L is film length, γ is IL surface tension, and κ is curvature of IL-vacuum interface at the conjunction of film and reservoir (Supplementary Table. 2, Supplementary Note 4, Supplementary Fig. 13). We depict the calculation results in Fig. 2h. Since the boundary slip length of IL-SiO₂ interface depends on the surface potential and ranges from 2 nm to over 16 nm⁵⁹, we employ the lower (2 nm) and upper (16 nm) limits of slip length to give an estimation on the possible range of flow speed in Fig. 2h as the blue shaded region, and the calculation result of average value 10 nm⁵⁸ is shown by the deep blue line.

The relationship between flow velocity and film length can also be measured experimentally. As shown in Supplementary Fig. 4, the HFIB scans the designed pattern row by row, so the beam speed vertical to the CTL can be calculated as:

$$v_{beam} = \frac{s_{vertical}}{N_{row} \cdot \tau + V_{refresh}}, \quad (5)$$

Where, $s_{vertical}$ is the vertical scan spacing, N_{row} is the number of scan spots in one row, τ is the dwell time that HFIB stay at a single spot, $V_{refresh}$ is a small time (10 μs) that NPVE takes to reset the HFIB for next row of scan. If the vertical speed of HFIB exceeds the film velocity, the distance between the scan spot and the liquid film CTL would increase until the scan spot is too far ahead of the film which would cease to flow. The critical interaction distance with given beam parameters can be determined experimentally (Supplementary Fig. 14a). Due to the pronounced impact of dose density on the flow velocity of liquid film (Supplementary Fig. 14b), the flow velocity measurements are conducted under the same dose density by keeping the beam current $I = 1 \text{ pA}$, scan spot spacing $s = 1 \text{ nm}$ and dwell time $\tau = 2 \text{ }\mu\text{s}$ constant and only alter the width of the rectangle pattern. The pattern width controls the N_{row} and consequently alters the beam vertical speed. By scanning rectangle patterns with same length but different widths outwards from the reservoir CTL, the vertical speed of beam can be changed at constant HFIB dose density. The IBFW film length decreases with the beam speed

increasing (Supplementary Fig. 14c), and consequently the average flow velocity of films with different length can be measured. The change width measurement results are shown in Fig. 2h by the hollow orange stars. The results of film speed at extremely long film lengths are acquired by first fabricating a long liquid film ($300\ \mu\text{m} \times 10\ \mu\text{m}$, $600\ \mu\text{m} \times 10\ \mu\text{m}$, and $900\ \mu\text{m} \times 10\ \mu\text{m}$ respectively) from the reservoir, then the change width measurements are conducted at the front of the long film. Since the fabrication of extreme long film can be time consuming, these data are only measured once. The other experiments are repeated for at least ten times with the average value and standard error shown in Fig. 2h.

The shortage of the change-width method is that the NPVE scan pattern assembling limits the maximum velocity the beam can move vertically. To overcome such limitation, we adopt single spot scan method. In which, a line pattern made up by a series of scan spots is used. The scan speed is altered by changing the vertical refresh time between each scan spot, while keeping beam current 1 pA, dwell time 10 μs and spacing 1 nm all constant. The dwell time is elongated to compensate the dose density reduction, since the scan area is influenced by neighboring scan spots in a rectangular pattern. All single spot measurements are repeated at least five times. The results of single spot scan are represented by the orange stars in Fig. 2h.”

(Page 7-8, Main text)

“Figure 2: (h) IBFW film flow speed as a function of liquid film length. The single spot experiments are conducted with a line pattern of scan spots, at constant beam current 1 pA, dwell time 10 μs , spacing 1 nm. The change width experiments are conducted with rectangular pattern of scan spots with constant length 20 μm and different width, while keeping the beam current beam current 1 pA, dwell time 2 μs , spacing 1 nm constant to keep the dose density unchanged. The blue shaded region is the range between the calculate velocity lower and upper limit due to the range of slip lengths^{58,59}, the deep blue line is the calculation result employing the average slip length⁵⁸.”

(Page 24, Main text)

“Supplementary Figure 14: Flow speed measurement experiments.

(a) The single spot scan style in HIM, which scans one-dimensionally along the pattern direction, is applied to analyze the interaction spatial range of a single irradiation spot quantitatively. The critical maximum spacing between neighbor scan spots to induce liquid flow, s_c , represents the upper limit for the interaction spatial range of the beam spot. When scan spot spacing exceed s_c , HFIB fails to induce continuous flow however large the dose is. The injection dose density, D , is regulated by changing beam current, I , at constant dwell time 100 μ s, or changing dwell time, τ , at constant beam current 0.7 pA. The relationship between s_c and D of each beam spot is plotted. (b) Maximal flow velocity as a function of the beam current I for a 20 μ m length rectangle pattern. The scan speed is changed by the pattern width. The flow speed increases with increasing beam current at same dwell time, 5 μ s, and spacing, 1 nm. (c) An example of the change width method for the measurements of flow velocity at different film length.”

(Page S31, Supplementary)

Comment #5: What are effects of temperature gradient and evaporation on the present technique?

Response: We thank the reviewer for the inspiring question. Temperature gradients can alter the flow of IL by Marangoni effect or evaporation-absorption effect. In our cases, the temperature gradient and evaporation induced by the HFIB are insignificant. We investigated the heating effect and the Marangoni effect of He ion beam on IL liquid by employing SRIM software as described in Supplementary Note 1. We find out the temperature difference induced by HFIB is less than 1 K, and the consequent surface tension gradient is not enough to induce liquid flow. Besides, the kinetic energy difference induced by this heating is $k_b \Delta T < 1K \cdot k_b = 8.62 \times 10^{-5} eV$. While the solvation energy of ion pairs in [EMIM][DCA] ionic liquid is 11.62 eV, which is the energy barrier for a single ion to overcome to evaporate from the liquid phase. Consequently, the temperature effect on the evaporation of ions from reservoir is negligible. Also, the ILs are known for their negligible vapor pressure (0.157 Pa at 499 K, the value at room temperature is negligible). Consequently, we believe the effect of temperature and evaporation are irrelevant.

Nevertheless, temperature is always an important parameter for a liquid film system, and the reviewer’s question inspires us to use external heating to explore intensive thermal influence.

The viscosity of ILs can be lowered by temperature increasing, which may improve the IBFW fabrication efficiency and help this technique be more efficient and practical. We thank the reviewer and we will settle the external heating tests in near future.

“We also exclude the potential roles played by HFIB induced surface morphological or chemical modification effects and heating effect induced Marangoni flow in Supplementary Note 1, Supplementary Fig. 16, and Supplementary Fig. 17. More details of MD can be found in Supplementary Note 6. **Since the temperature increasing effect is less than 1 K based on our calculation. The thermal energy difference, $k_b\Delta T < 1K \cdot k_b = 8.62 \times 10^{-5}eV$, induced by HFIB is far less than the energy barrier for ions to overcome to evaporate from the liquid phase. Therefore, the evaporation effect may also be insignificant in current experiments.**”

(Page 8, Main text)

Responses to Reviewer #2

This manuscript describes experiments demonstrating a novel way to print ionic liquids on insulating substrates using a He-ion beam writing. To my knowledge results are novel, their importance to the field of controlling liquid flows are hard for me to judge. The explanation of the working principle has several apparent deficiencies.

There are several remarks I would like to make:

We thank the reviewer for the comments.

Comment #1: *-The relevance for applications was not entirely clear to me. Several references were cited in the beginning, listing a broad range of applications for “programmable control of fluid motion”. But as I understood, the present technique works for ionic liquids only. Thus, I was left wondering what the applications are specific to ionic liquids. That needs to be explained much better.*

Response: We thank the reviewer for the very important suggestion. Accordingly, we have analyzed two specific application fields and carried out demos. First, this Ion Beam induced Film Wetting (IBFW) technology can be useful for the miniaturization of chemical reaction chips. The ionic liquids (ILs) film pattern can serve as the micro flow channel for the aqueous or organic solutions of analytes injected into the IL droplet reservoir since ILs are good solvent for water and organic solutions. With the reagents dissolve and diffuse into the film channel, in-situ chemical reaction can be conducted. Second, IBFW technology can be combined with electrochemistry to transform patterned liquid film into solid materials. The ILs known for wide potential window, low toxicity and thermal stability, can be used as solutions for electrochemical experiments. With patterned liquid film printed to electrode surface, solid materials can be deposited with designed patterns through electrochemical procedures, which is important for nano-circuit manufacture. Adjustments have been made to our manuscript and are listed here.

1. We modified the introduction to introduce potential application fields suitable for ionic liquids.
2. We added a new section ‘Applications of IBFW’ for further demonstration. We used IBFW to fabricate liquid channels as a guidance for aqueous solutions of reagent which were later injected into the IL droplets for the in-situ colorimetric reactions of SCN^- with various metal ions. This experiment demonstrates that IBFW liquid film pattern can act as a stable flow channel for later injection of analyte solutions.
3. We conducted electrochemical experiments to reveal that IBFW is also capable of transforming liquid film pattern into desired solid materials. Electrochemistry analytes are dissolved in 1-ethyl-3-methylimidazolium bis(trifluoromethylsulfonyl)imide ([EMIM][NTf₂]). The solution droplet was settled on ultra-thin metal film (15 nm) deposited on SiO₂ wafer, and film patterns were printed by IBFW. The following electrochemical deposition transforms the liquid pattern into solid materials. Since the stopping range of 30 kV Helium ions exceeds 290 nm, a 15 nm metal film exerts no significant influence on the surface charge injection of HFIB, and consequently the IBFW

can be conducted on very thin metal film. Monte Carlo simulation results were added to explain the penetrating capability of He ions (Supplementary Fig. 9b). We also discussed the potential influence of electrowetting on the IBFW film, and drew the conclusion that the voltage we adopted won't overcome the contact angle hysteresis and the IBFW films were not influenced by the deposition voltage.

For now, we only tested Ag nano-particles and AgTCNQ (7,7,8,8-tetracyanoquinodimethane) complex, but more electrochemistry tests can be undertaken to test the capability of solid deposition of both organic and inorganic materials.

“Based on the IBFW inducing mode, we develop a nano-printing technique of ILs, with film thickness down to 20~30 nm, minimal line width about 100 nm and corner radius down to 20 nm, and compare its performances with the reported methods. Besides, ILs are also known for their unique properties such as wide electrochemical potential window, high ionic conductivity, low toxicity and thermostability. These features make ILs increasingly important as electrolytes for lithium battery^{40,41} and electrodepositions of various materials ranging from metal nanoparticles^{42,43}, metal organic complexes⁴⁴ to conducting polymer films⁴⁵. We further demonstrate the IBFW as a versatile tool for various application fields including gas sensing circuit, in-situ chemical reaction chip, and electrochemical deposition of solid materials with desired patterns. The simplicity and versatility of IBFW technique suggests prospect in a range of liquid manipulation applications. By combining with electrochemical procedures, such technique can not only produce patterned liquid film but also solid materials which reveals possibility in nano-transistors fabrication⁴⁶, energy devices⁴⁰ and immunosensor circuit printing⁴⁷. We expect this technique can open a new avenue for applications in nano-printing and nano-circuit manufacturing.”

(Page 2-3, Main text)

“Applications of IBFW

As discussed previously, IL film pattern prepared by IBFW technology manifests three distinguishable features. First, the ultralow film thickness down to 30 nm indicates a high surface-volume ratio which is a key role in improving the gas sensing circuit sensitivity. Second, the capability of fabricating liquid film with desired pattern in a programmable and rewritable manner, which is important for in-situ chemical reaction and microfluidics chips. Third, the ILs are widely used in electrochemistry and reveal the possibility of transforming liquid film pattern into various solid materials ranges from organic to inorganic compounds.”

(Page 9, Main text)

“To demonstrate the potential of IBFW for microfluidics chip fabrication, we design an in-situ chemical reaction micro-chip. In Fig. 4c, the schematics shows a crosshair shaped micro fluid channel connects four separated droplets in four directions. Four square expansion windows are made on each part of the channel for the convenience of observation. The top-left inset shows the chip with four droplets on finger-tip. After the IBFW fabrications, 0.1 μL of sodium thiocyanate solution (NaSCN 0.1 M in deionized water) is injected into the top droplet, and serves as the colorimetric reagent for the detection of and in-situ reaction with different metal ions. After the injection of NaSCN, 0.1 μL of 0.1 mM Fe³⁺ solution, 0.1 mM Cu²⁺ solution and

0.1mM Co^{2+} are injected into the bottom, left and right droplets respectively. The microchip is rested in atmosphere for 20 min for the metal ions fully diffuse into the channels and react with SCN^- within different square windows with the ion names printed previously. Then the sample is transferred into vacuum chamber for 48 h to diminish the water content in the solution system, which will alter the hydration status of the metal ion complexes and improve the colorimetric visibility. The red complex $\text{Fe}(\text{SCN})_3$ deposits in the bottom window. The gray deposition in the left window is complex $\text{Cu}(\text{SCN})_2$. And the blue deposition in the right window is complex $\text{Co}(\text{SCN})_2$. As shown in Fig. 4c, the SCN^- participates into 3 different reactions within several hundreds of micrometers flow channel. The time series pictures are shown in Supplementary Fig. 20, and the liquid film patterns remain unchanged during the experiments which last for over one month. The IBFW fluid channel exhibits great stabilities against vacuum/air transferring, the injection of solutions into droplet reservoir, and gravity. Such behavior demonstrates the robustness of the IBFW liquid film. **More importantly, all reagents are dissolved in deionized water then injected into the IL droplets and diffuse into the IL flow channel. This experiment demonstrates that IBFW liquid film pattern can act as a stable flow channel for later injection of analyte dissolved in water, ethanol and various molecular liquids due to the amphiphilicity of ILs. Such results greatly broaden the potential application fields for IBFW.”**

(Page 10, Main text)

Figure. 4: (c) In-situ chemical reaction chip. The left part is the schematics of the chip. Crosshair channels connect four droplets which are used to inject reagent water solutions. 0.1 μL of 0.1 M NaSCN, 0.1 mM Co^{2+} , 0.1 mM Fe^{3+} and 0.1mM Cu^{2+} are injected clockwise into four droplets (top, right, bottom, left). The inset on the top-left compares the size of microchip with fingertip, with red circle shows the four droplets. The right part is the optical image of the reaction chip that has been stored in vacuum chamber for 48 h after the injection, the color of complexes is more obvious with water removed from the system.

(Page 27-28, Main text)

Supplementary Figure 20: Time series images of in-situ chemical reaction chip.

(a) HIM image of channels right after the IBFW fabrication. (b) The fluid channel first transferred to the optical microscope; the colorless transparent fluid channel is a bit hard to be distinguished from the transparent silica substrate. (c) After the injection of all solutions and reacts in atmosphere for 5 min. (d) The whole picture of the reaction chip before transferred to vacuum chamber. (e) 48 h after metal ions injection and storing in vacuum. After stored in vacuum chamber, the water has been eliminated, and the color of the different complexes become obvious. The sample goes through several times of transferring between vacuum chamber and air environment and the injection of analyte solutions into droplet reservoirs, while the liquid film pattern remains barely changed through the experiments. Such results demonstrate the stability of IBFW liquid film channel to the exposure of air environment and to non-direct physical contact for solution injections. (f) The reaction chip one month after injection (one week for gravity stability test). The chip is mounted on a customized sample holder which can adjust the tilting angle from 0-180° to test the IBFW film stability against gravity. The inset shows the mounted sample with tilting angle 120° and is stored for one week.

(Page S37, Supplementary)

“Due to their unique properties, ILs have been proved to be an important category of solvent and electrolytes. Here we demonstrate that, by further combining with electrochemical procedure, the IBFW also manifests the capability of transforming liquid film pattern into various solid materials. Fig. 4d shows the schematics of a three-electrode electrochemistry experiment. A droplet of 1-ethyl-3-methylimidazolium bis(trifluoromethylsulfonyl)imide ([EMIM][NTf₂]) serves as the solvent of possible analytes, which are Silver bis(trifluoromethylsulfonyl)imide (Ag[NTf₂]) or a mixture of Ag⁺ and 7,7,8,8-tetracyanoquinodimethane (TCNQ) in our experiments. The liquid film pattern is fabricated by the IBFW technology onto the thin gold electrode and would be subsequently transformed into

nanoparticles. A 10 nm Au and 5 nm Ti film deposited onto SiO₂ serves as the working electrode and is connected to the workstation by a Pt probe. As revealed by the MC simulation results (Supplementary Fig. 9 b), the He ions with vertical stopping-range exceeds 290 nm can easily penetrate the 15 nm metal film and deposit positive charges into the 300 nm SiO₂ layer underneath. Therefore, the IBFW can be achieved on an ultra-thin metal film deposited on insulating substrate, and is not contradict with the conclusion that pure conducting substrates lead to the failure of IBFW shown in Fig. 2f. The counter electrode is a Pt probe emerged in IL, and a silver-plated probe serves as a pseudo reference electrode.

In Fig. 4e, we present an example of solid particles deposited from IBFW liquid film. The upper part shows silver nanoparticles with designed pattern (a 20 μm × 3 μm channel and PKU letters) potentiostatically deposited at -0.2 V (vs. Ag) for 180 s onto the gold electrode surface. And the lower part shows blue AgTCNQ particles on gold surface make up a 50 μm × 10 μm rectangular pattern deposited at -0.1 V (vs. Ag). Noteworthy, the electrowetting phenomenon is ubiquitous in IL-electrode systems with contact angle hysteresis ranges from several to tens of degrees⁶³. When the voltage applied between the IL and electrode surface is large enough to overcome contact angle hysteresis, the contact line of the IL droplet will be shifted and the IBFW patterns will be jeopardized. According to Liu et.al.⁶⁴, a negative bias voltage exceeds -1 V would induce significant contact angle decreases for [EMIM][NTf₂] on gold with contact line spreading forwards, which is also observed in our experiments. A negative voltage smaller than -0.5 V, on the contrary, does not influence the contact angle obviously. Therefore, the deposition voltages adopted in current work does not influence the contact line position or the IBFW liquid film.

Four cycles of cyclic voltammogram (CV) of 10 mM Ag⁺ in IL is shown in Fig. 4f. The reduction peak of Ag⁺ takes place at -0.29 V (vs. Ag) with the peak current decreases as the scan cycles increases. We believe that the micro litter droplet with limited analyte dissolved lead to such results. The Ag⁺ concentration decreases quickly after each cycle of CV scan, and the electron transfer is slower for the oxidation of Ag metal. In Fig. 4g, we test the CV curves of five scan cycles of IL droplet with 6 mM Ag⁺ and 5 mM TCNQ. Two reduction peaks can be distinguished, one at 0.095 V (vs. Ag), and the other at -0.21 V (vs. Ag). The first peak corresponds to the reduction of Ag⁺ to Ag⁰ (metal)⁴⁴, and the second peak is related to the formation of AgTCNQ complex (solid). Finally, we conceptually validate that IBFW technology is capable of transferring liquid film pattern into various solid materials and reveal the possibilities can be produced by combining IBFW with electrochemical procedures. The three-electrode experiment configuration, example of open-circuit-potential V-t curve and potentiostatic deposit I-t curve are shown in Supplementary Fig. 21.”

(Page 10-11, Main text)

“As summarized in Fig. 4h, the IBFW technology manifests several intriguing features that can be harnessed for a variety of application fields. The IBFW technique can fabricate patterned ILs film with 30 nm thickness, 100 nm spatial resolution and over hundreds of μm film length on insulating substrates (or coated with conducting metal films with 10¹ nm thickness). The surface-volume ratio endowed by the nanometer scale thickness can largely enhance the sensitivity of the IBFW film, and can be utilized in gas sensing circuit. The good solubility and biocompatibility of ILs make them suitable for the dissolve of various analytes.

The IBFW film also exhibits robustness against air environment exposure, gravity, and physical contact to droplet reservoir. Such features reveal that the IBFW film can act as stable flow channel for the analyte solutions injected to the reservoir, and can largely simplify the fabrication procedures of micro/nanofluidic chips. Last but not least, IBFW liquid film with wide potential window can be combined with electrochemical procedures and the patterned liquid film can be transformed into different solid particles. Such results demonstrate the IBFW as a versatile tool for both nanofluidics and liquid/solid materials printing.”

(Page 11, Main text)

Figure 4 (d) The schematics of a three-electrode electrochemistry experiment. A droplet of IL with IBFW liquid film pattern printed onto the working electrode is the solvent for analytes.

10 nm Au and 5 nm Ti deposited to 300 nm TOX SiO₂ serves as the working electrode, Pt probe and silver-plated probe stuck into the droplet are the counter and pseudo-reference electrodes respectively. **(e)** An example of the patterned solid particles deposited on Au electrode surface. The upper part shows Ag particles make up a 20 μm × 3 μm rectangular film with PKU letters pattern which are deposited to Au surface at -0.2 V (vs. Ag). The lower part shows blue AgTCNQ particles make up a 50 μm × 10 μm pattern which are deposited to Au surface at -0.1 V (vs. Ag). Both are potentiostatically deposited for 180 s. **(f)** Four cycles of cyclic voltammogram of 10 mM Ag⁺([EMIM][NTf₂]) solution. The reduction peak at -0.295 V (vs. Ag) can be seen, with peak current decreases with cycles. After four cycles, the reduction peak of Ag⁺ to Ag metal becomes less obvious. **(g)** Five cycles of cyclic voltammogram of 6 mM Ag⁺ and 5 mM TCNQ ([EMIM][NTf₂]) solution. The first reduction peak of Ag⁺ to Ag metal at 0.095 V (vs. Ag) can be seen. The +300 mV shift of Ag reduction peak at present of TCNQ is consistent with literature⁴⁴. The reduction peak current of Ag reduces quickly and become hard to distinguish as cycle increases. **(h)** A summary schematic to show the distinct features of IBFW technique, and the potential application fields that are suitable for IBFW.

(Page 27-28, Main text)

Supplementary Figure 21: The electrodeposition experiments of AgTCNQ.

(a) The three-electrode configuration adopted in current work. **(b)** The green line is the V-t curve of AgTCNQ open circuit potential measurement, which serves as a reference for the subsequent experiments. The purple line is the deposition current vs. time curve of AgTCNQ deposition at constant potential, -0.1 V (vs. Ag).

(Page S38, Supplementary)

Comment #2: *The fabrication has to be performed in an ultra-high vacuum environment. It was not very clearly stated or studied, what happens to the samples after they are taken out of the HIM and exposed to air, tilting of the chip (gravity pulling in another direction), static charges, mechanical contact or other environmental variables. Common sense picture of liquids gives the impression that the “circuit” is not very stable. If it is, data needs to be provided to convince the reader otherwise.*

Response:

We thank the reviewer for the inspiring suggestions.

IBFW films are stable to the exposure to air environment or transferring between vacuum/air conditions. The in-situ reaction chip presented in Supplementary Fig. 20 proves that the IBFW liquid film is quite stable after exposure to atmosphere, and the transfer between vacuum chamber and air environment repeatedly. The negligible vapor pressure of ionic liquids accounts for its vacuum/air environment stability. The evaporation of ionic liquids at ambient condition is indiscernible and do not suffer from coffee ring effect. The contact line of ionic liquid is pinned by solid substrate and the sample patterns are barely changed for one month in Supplementary Fig. 20.

IBFW films are stable to gravity. We employed a customized sample holder which can adjust the tilting angle from 0 to 180°. The in-situ reaction chip was mounted to the holder with tilting angle settled at 120° (inset of Supplementary Fig. 20f), and the reaction chip with IBFW patterns was stored in the sample box for one week. The optical images (Supplementary Fig. 20) show that the IBFW film pattern was barely changed through the experiments. The micro-nano meter size of liquid film can explain this stability. For liquid film down to nm thickness and micrometer length, the surface tension, disjoining pressure far exceeds the gravity, so turning the chip upside down or tilting dose not influence the film pattern.

IBFW films are relatively stable at mild static charge conditions. In the electrochemistry measurement, solid patterns were deposited from the liquid film at a relative low bias voltage (-0.1 ~ -0.2 V) see Fig. 4e. Such a result indicates that the IBFW film exhibits relative stability to static charge. The electrowetting phenomenon is ubiquitous in IL-electrode systems with contact angle hysteresis ranges from several to tens of degrees (Ref. [63]). When the voltage applied between the IL and electrode surface is large enough to overcome contact angle hysteresis, the contact line of the IL droplet will be shifted and the IBFW patterns will be jeopardized. According to Liu et.al. (Ref. [64]), a negative bias voltage exceeds -1 V would induce significant contact angle decreases for [EMIM][NTf₂] on gold with contact line spreading forwards. Such phenomenon is also observed in our cyclic voltammogram (CV) experiments, where the entire contact line was moved and the IBFW film patterns were destroyed. A negative voltage smaller than -0.5 V, on the contrary, does not influence the contact angle obviously. Therefore, the deposition voltages adopted in current work does not influence the contact line position or the IBFW liquid film.

IBFW film cannot remain the same after a direct physical contact of daily subjects, but direct contact to the droplet reservoir does not change the liquid film. In the in-situ reaction chip experiments and the electrochemical experiments, the droplet connected to the liquid film is penetrated by micropump for solution injection or electrode probes. The liquid film remains

intact through the experiments, and remains barely changed for days. As can be seen from the comparison of microchip optical images and HIM images before and after the injection and reaction (Supplementary Fig. 20).

IBFW film is vulnerable to a direct exposure to dust or other contaminations. Since the size of the liquid film is small, too much dusts absorbed to the chip surface or the liquid will damage the liquid film. Short time exposure to the atmosphere condition is acceptable. The IBFW samples are stored in a vacuum chamber and are only took out for experiments.

“As shown in Fig. 4c, the SCN^- participates into 3 different reactions within several hundreds of micrometers flow channel. The time series pictures are shown in Supplementary Fig. 20, and the liquid film patterns remain unchanged during the experiments which last for over one month. The IBFW fluid channel exhibits great stabilities against vacuum/air transferring, the injection of solutions into droplet reservoir, and gravity. Such behavior demonstrates the robustness of the IBFW liquid film. More importantly, all reagents are dissolved in deionized water then injected into the IL droplets and diffuse into the IL flow channel. These experiments demonstrate that IBFW liquid film pattern can act as a stable flow channel for later injection of analytes dissolved in water, ethanol, and various molecular liquids due to the amphiphilicity of ILs.”

(Page 10, Main text)

Supplementary Figure 20: Time series images of in-situ chemical reaction chip.

(a) HIM image of channels right after the IBFW fabrication. (b) The fluid channel first transferred to the optical microscope; the colorless transparent fluid channel is a bit hard to be distinguished from the transparent silica substrate. (c) After the injection of all solutions and reacts in atmosphere for 5 min. (d) The whole picture of the reaction chip before transferred to vacuum chamber. (e) 48 h after metal ions injection and storing in vacuum. After stored in vacuum chamber, the water has been eliminated, and the color of the different complexes

become obvious. The sample goes through several times of transferring between vacuum chamber and air environment and the injection of analyte solutions into droplet reservoirs, while the liquid film pattern remains barely changed through the experiments. Such results demonstrate the stability of IBFW liquid film channel to the exposure of air environment and to non-direct physical contact for solution injections. **(f) The reaction chip one month after injection (with one week for gravity stability test).** The chip is mounted on a customized sample holder which can adjust the tilting angle from 0-180° to test the IBFW film stability against gravity. The inset shows the mounted sample with tilting angle 120° and is stored for one week. **(Page S37, Supplementary)**

“We delineate a system free energy ratio scenario for a liquid film system with unit length/width and thickness vary from 1 nm to 4 mm in Fig. 3e, and the film thickness and corner radius are compared with published results^{16,18,23,24}. **System free energy composes of the volumetric term (gravity, electrostatic, etc.), the surface tension term, and the disjoining pressure term.** The surface tension remains fixed magnitude of 10^1 mN/m, and its contribution to the system is almost constant with the thickness variation. When system size exceeds capillary length, $\lambda_{capillary}$, the volumetric term contributes most of the system free energy. **For any system with characteristic length below capillary length, however, the influence of volumetric term (gravity etc.) can be neglected.** At millimeter to micrometer range, the surface tension dominates, and the majority of traditional microfluidics methods belong to such region, with the spatial resolution difficult to approach nanoscale.”

(Page 9, Main text)

“In Fig. 4e, we present an example of solid particles deposited from IBFW liquid film. The upper part shows silver nanoparticles with designed pattern (a $20 \mu\text{m} \times 3 \mu\text{m}$ channel and PKU letters) potentiostatically deposited at -0.2 V (vs. Ag) for 180 s onto the gold electrode surface. And the lower part shows blue AgTCNQ particles on gold surface make up a $50 \mu\text{m} \times 10 \mu\text{m}$ rectangular pattern deposited at -0.1 V (vs. Ag). Noteworthily, **the electrowetting phenomenon is ubiquitous in IL-electrode systems** with contact angle hysteresis ranges from several to tens of degrees⁶³. **When the voltage applied between the IL and electrode surface is large enough to overcome contact angle hysteresis, the contact line of the IL droplet will be shifted and the IBFW patterns will be jeopardized.** According to Liu et.al.⁶⁴, a negative bias voltage exceeds -1 V would induce significant contact angle decreases for [EMIM][NTf₂] on gold with contact line spreading forwards, which is also observed in our experiments. **A negative voltage smaller than -0.5 V, on the contrary, does not influence the contact angle obviously. Therefore, the deposition voltages adopted in current work does not influence the contact line position or the IBFW liquid film.**”

(Page 11, Main text)

Comment #3: *Some data was provided with different substrates. What about varying the surface treatment, or cleaning? How sensitive is the success of the writing to the cleanliness of the substrate? You say you anneal the substrates at 320 C in hydrogen flow. Is that to remove organic contaminants? What if you skip the annealing step? Or use oxygen plasma cleaning*

instead?

Response:

We thank the reviewer for the questions. Vary the surface treatment or cleaning procedures does not influence the IBFW experiments as long as the surface cleanliness is guaranteed. The cleanliness of the substrate is crucial for the IBFW experiment. The liquid film is about 30 nm thick, so any dust or pollutants can easily influence the liquid film. Besides, the Helium ion microscope (HIM) is sensitive to organic pollutants. Therefore, the solid substrate and liquid have to be cleaned thoroughly before transferred into the HIM chamber, to eliminate potential volatile pollutants. The annealing progress is to remove possible organic contaminants under this consideration. Skipping the annealing or other cleaning process may do harm to the instrument, so we did not attempt to do so. We tested with a SUNJUNE oxygen plasma cleaner and achieved the same cleanliness we need and included it into the Methods section.

“IBFW experiment

The fabrication details of the experiments used SiO₂ wafers are described below. An amorphous SiO₂ layer with a thickness of approximately 3 μm is deposited by plasma-enhanced chemical vapor deposition (PECVD) on a 500 μm thickness quartz substrate surface. Prior to the experiment, chips are cleaned with acetone, ethanol, and ultrapure water sequentially with ultrasonication. To remove any surface organic pollutants, **the wafers are annealed in a tubing furnace at 320 °C with a mixture gas flow of hydrogen and argon for at least 3 hours, a SUNJUNE oxygen plasma cleaner can also fulfill the purpose.**”

(Page 11, Main text)

Comment #4: *How do you know that the ions of the ionic liquid stay intact (are not broken down) under the He ion irradiation? Typically, organic compounds (such as resist for example) can be strongly modified (bonds broken etc) with high energy ion beams.*

Response:

We thank the reviewer for the important comments. Both ions (EMIM⁺ and DCA⁻) employed are reported to show good radiolytic stability. The imidazolium cations have shown good radiolytic stability due to the aromatic ring can adsorb and relax the radiation energy. The dosage of helium focused ion beam (HFIB) in our IBFW experiments is much smaller than the damage dosage reported in literature^{1,2}(main text Ref. [79, 80]). The radiolytic stability of imidazolium based ionic liquids under He²⁺ radiation was tested¹. Decompose products H₂ were measured by NMR with irradiation dose ranges from 2 to 400 kGy, and no trace of reaction (< 1%) was detected for 2 kGy irradiation. The irradiation dosage of Helium ion is 0.25 Gy during a typical IBFW experiment. For a 10 μm × 1 μm film pattern with scanned area *A* fabricated from a 1 μL IL droplet with 1 pA beam current *I*, 1 μs dwell time *τ*, 1 nm scan spacing *s*, and 30 kV accelerate voltage *U*, the dosage can be calculated as:

$$D = \frac{\text{Incident Energy}}{\text{Mass of IL}} = \frac{U \cdot I \cdot \tau \cdot \frac{A}{s^2}}{V \cdot \rho} = \frac{30 \times 10^3 \text{V} \times \frac{10^{-12} \text{C/s}}{1.602 \times 10^{-19} \text{C/e}} \times \frac{10 \times 1 \mu\text{m}^2}{1 \times 1 \text{nm}^2}}{0.001 \text{cm}^3 \times 1.11 \text{g/cm}^3} = 0.25 \frac{\text{J}}{\text{kg}} = 0.25 \text{ Gy}.$$

The [EMIM⁺] adopted in IBFW has a shorter alkyl chain attached to the aromatic ring, so the radiolytic stability of [EMIM⁺] should be better than the [BMIM⁺] tested in literature. Therefore, we believe that the cations of imidazolium are intact.

Anions structures also influence the radiolytic stability. The [DCA⁻] is the most radiolytic stable anions among all anions². The stability of a free radical decreases when the hybridization of the carbon goes from sp³ to sp² to sp. Thus, it is hard to induce cleavage of the C≡N bond of [DCA⁻] to form ·C=N.

Besides, the electrochemical results of the Ag(NTF₂) and TCNQ were consistent with literatures (main text Ref. [43, 44]), we believe they are not infected by the HFIB irradiation. In summary, the radiolysis effect on our sample can be negligible.

We plan to conduct a XPS or Raman experiment to verify our idea, but the instrument is not available at the time, we shall conduct it as soon as possible.

1. Allen, D. *et al.* An investigation of the radiochemical stability of ionic liquids. *Green Chem.* **4**, 152–158 (2002).
2. Xue, Z., Qin, L., Jiang, J., Mu, T. & Gao, G. Thermal, electrochemical and radiolytic stabilities of ionic liquids. *Phys. Chem. Chem. Phys.* **20**, 8382–8402 (2018).

“Supplementary Note 7: Influence of radiolysis effect in IBFW

The ILs and the electrochemistry analytes in the main text remain intact through the IBFW experiments. The conclusion is based on the following discussion.

The imidazolium cations have shown good radiolytic stability. The dosage of HIB in our IBFW experiments is much smaller than the damage dosage reported in literature^{30,31}. The radiolytic stability of imidazolium based ionic liquids under He²⁺ radiation were tested¹. Decompose products H₂ were measured by NMR with irradiation dose ranges from 2 to 400 kGy, and no trace of reaction (<1%) was detected for 2 kGy irradiation. The irradiation dosage of Helium ion is 0.25 Gy during a typical IBFW experiment. For a 10 μm × 1 μm film pattern with scanned area *A* fabricated from a 1 μL IL droplet with 1 pA beam current *I*, 1 μs dwell time *τ*, 1 nm scan spacing *s*, and 30 kV accelerate voltage *U*, the dosage can be calculated as:

$$D = \frac{\text{Incident Energy}}{\text{Mass of IL}} = \frac{U \cdot I \cdot \tau \cdot \frac{A}{s^2}}{V \cdot \rho} = \frac{30 \times 10^3 \text{V} \times \frac{10^{-12} \text{C/s}}{1.602 \times 10^{-19} \text{C/e}} \times \frac{10 \times 1 \mu\text{m}^2}{1 \times 1 \text{nm}^2}}{0.001 \text{cm}^3 \times 1.11 \text{g/cm}^3} = 0.25 \frac{\text{J}}{\text{kg}} = 0.25 \text{ Gy}.$$

The [EMIM⁺] adopted in IBFW has a shorter alkyl chain attached to the aromatic ring, so the radiolytic stability of [EMIM⁺] should be better than the [BMIM⁺] tested in literature. Therefore, we believe that the cations of imidazolium are intact.

Anions structures also influence the radiolytic stability. The [DCA⁻] is the most radiolytic stable anions among all anions³¹. The stability of a free radical decreases when the hybridization of the carbon goes from sp³ to sp² to sp. Thus, it is hard to induce cleavage of the C≡N bond of [DCA⁻] to form ·C=N.

Besides, the electrochemical results of the Ag(NTF₂) and TCNQ were consistent with literatures (main text Ref. [43, 44]), we believe they are not infected by the HFIB irradiation. In summary, the radiolysis effect on our sample can be negligible.”

(Page S16, Supplementary)

Comment #5: *What sets the “minimal line width “ scale of 100 nm? Shouldn ’ t it in principle*

be much narrower, as the He ion beam interaction volume at the surface is very small (Fig S7) (hence its high imaging resolution < nm).

Response:

We thank the reviewer for the important comment. The 100 nm resolution of liquid film is determined by the surface charge injected by the helium focused ion beam (HFIB). The beam spot of HFIB is 0.5 nm, but the He ions interact with the sample atoms and would be diffracted while traveling through the solid materials. Our Monte Carlo simulation results (Supplementary Fig. 9) and Ohya et. al's results (main text Ref. [48]) revealed the Helium ions irradiating SiO₂ (we also tested 15 nm metal on 300 nm SiO₂ for electrochemistry experiments) substrates have lateral projection length ranges from 90 nm to 110 nm, therefore the positive charges distribute around 200 nm in lateral direction for single injection spot of He ions. The 100 nm range is probably the center region of surface charging that is strong enough to induce ion emission and form liquid film. Consequently, the resolution of IBFW liquid film is 100 nm for the amorphous SiO₂ substrate in our experiments. Since electron beams can be diffracted more easily than HFIB, the surface charging region induced by electron beam is much larger. As a result, 5 kV electron beam fails to achieve the programmable control of the liquid film. We conducted experiment employing scanning electron microscope (SEM) for comparison. We added an explanation to liquid film resolution and a comparison of the liquid inducing results of HFIB and electron beam in the main text and Supplementary Fig. 8.

“When HFIB irradiates at or near the CTL of droplet, the Helium ions generate special charges distribution⁴⁸ in the SiO₂ substrate, the positive surface charges induce the primary ion emission from the IL reservoir (Fig.2a). HFIB exhibits two distinguishable features compared with electron beam and other ion beams⁴⁸. **Firstly, the divergence of HFIB interacting with solid samples is much smaller than electron beam due to its larger mass⁴⁹, which lead to a more localized surface charging area and consequently a programmable control over liquid flow (Supplementary Fig. 8). Secondly, HFIB tends to penetrate sample and induce less damaging compared with Gallium beams⁴⁸.** The penetration depth exerts significant influence on IBFW's application potential in electrochemistry field. For example, 30 keV He beams with hundreds nm stopping range can easily penetrate a 10 nm Au and 5 nm Ti electrode deposited on SiO₂ wafer and induce patterned liquid flow without devastating effect to Au surface (Supplementary Fig. 9), while 30 keV Ga with stopping range less than 20 nm⁴⁸ is hard to penetrate metal films and can easily cause damage to the electrode.”

(Page 4, Main text)

“**Monte Carlo simulation results⁴⁸ reveal that the lateral projection length of 30 kV He ions irradiated on SiO₂ (also SiO₂ with 10 nm Au and 5 nm Ti layers) ranges from 90 nm to 110 nm (Supplementary Fig. 9a,b). The consequent positive charges (ions, holes) distribute over 10² nm in lateral direction and determine the ion emission and IBFW spatial resolution.**”

(Page 4, Main text)

Supplementary Figure 8: SEM images of liquid film induced by ion and electron beams.

(a) 30 keV helium focused ion beam induced liquid film on a SiO₂ substrate; (b) 5 keV electron beam induced liquid flow on the identical substrate. The liquid can also be induced to flow from the reservoir, but the flow pattern is out of control. The much more diffractive nature of the electron beam-sample interaction, and the consequent wider surface charges lateral distribution give rise to such results.

(Page S24, Supplementary)

Supplementary Figure 9: The HFIB irradiation interaction with liquid and substrates under experimental conditions. (a) Monte Carlo simulation results of 3000 He ion irradiating a ionic liquid film of 30 nm thickness above silica substrate, the accelerate voltage is 30 keV to reproduce the experimental conditions. The right part is the magnification of He ions' near liquid trajectory lines and spatial distribution. (b) the MC simulation results of He ions with 30 KV irradiate 10 nm Au, 5 nm Ti and 300 nm SiO₂. Similar to the results in the previous case, the HFIB can easily penetrate the first 15 nm layers of metal and deposit positive ions into the insulating SiO₂ substrate. Such results indicate that IBFW can also be done on insulating substrates with thin film of metal deposited on the surface. The vertical stopping range of He ions exceed 290 nm on both substrates. The lateral projection distances of He ions in both cases range from 90 nm to 110 nm in separated repeated simulations, which agree with the IBFW film resolution 100 nm. (c) the XPS spectra of the same substrate before (black) and after (red)

the irradiation of HFIB under experimental conditions with insets show the ESEM results of pristine and irradiated area of solid surface.

(Page S25-26, Supplementary)

Comment #6: *Related to that, I do not understand the suggested mechanism. In Fig. 2A the local surface charges, induced by the beam current, are located directly under the IL droplet. Yet, in the unnumbered Figure in Note S3 (Electric field driven ion emission calculation), the surface charge is located some distance away from the IL reservoir. Something doesn't match.*

Response:

We thank the reviewer for the comment. The two figures illustrate different situations and lead to this mismatch. The IBFW effect can be achieved by set the start scan position right at or near the contact line. As long as the surface charges injected by HFIB is strong enough to induce ion emission and form precursor film, liquid film can be induced. Fig. 2a shows the typical condition in standard IBFW experiments, where HFIB scans right at the contact line to achieve the best patterning performances. The unnumbered figure has been moved to the main text according to Comment #10 and labeled as Fig. 2d. When the start scan position is a bit away from the contact line, the surface charge injected by HFIB is still enough to induce liquid film, but the HFIB dosage required increases with this distance. Fig. 2d illustrates the mechanical balance between the surface tension and the electrostatic force exerted by surface charges located at a distance away from the IL-vacuum interface.

We have reconstructed the Working principal section, and a more detailed response can be found in the latter response to comments 7-10.

Figure 2: Working principle of IBFW nano-printing. (a-c) Schematic of the IBFW working principle. (a) When HFIB irradiates the CTL, the positive surface charging induces the primary anion emission. (b) When HFIB ceases to scan, surface charging dissipates and the emitted anions induce the secondary cation emission. (c) The emitted ions from previous stages form an ultra-thin precursor film, and the consequent disjoining pressure propels and stabilizes liquid film. (d) The mechanical balance between the surface charge density induced electrostatic force and surface tension of IL-vacuum interface. The distance between HFIB scans spot and contact line is d ; the surface charging uniformly distributes over a region with length scale, l_0 ; the surface-charge-exert electrostatic force and distorts the IL-vacuum interface, balanced by the surface tension γ . (e) Experimental HFIB dosage density to induce IL flow (ions per square of nm) and calculation results of critical dose density to induce significant ion emission. The inset shows when the scan position is separated too far from the CTL, the consecutive liquid film degenerates to the local protrusion flow as shown in Fig. 1b. (f) IBFW induced liquid film lengths (μm) on substrates with different conductivities. The right ordinate represents the overall ion emission number calculated by the beam parameters and sample characters. (g) Molecular dynamics simulation of [EMIM][DCA] droplet (640 ion pairs) deposited on fused silica substrate going through surface charges injection and removal. The arrows indicate the most directed movements of ions: the pale blue arrows at the beginning stage represent the surface charge induced primary anion emission; the purple arrows of cations represent the emitted anions induced secondary cation emission. The shaded regions (red) represent the surface injection region with positive charges. (h) IBFW film flow speed as a function of liquid film length. The single spot experiments are conducted with a line pattern of scan spots, at

constant beam current 1 pA, dwell time 10 μ s, spacing 1 nm. The change width experiments are conducted with rectangular pattern of scan spots with constant length 20 μ m and different width, while keeping the beam current beam current 1 pA, dwell time 2 μ s, spacing 1 nm constant to keep the dose density unchanged. The blue shaded region is the range between the calculate velocity lower and upper limit. The discrepancy at extreme long film length may be due to the HFIB irradiation history. The fabrication of film with hundreds μ m length usually takes hours of HFIB irradiation. The accumulated positive charges lead to a higher surface potential and a boundary slip length that exceeds the upper limit in literature, which give rise to an unexpected higher film speed.

(Page 22, Main text)

Comment #7: *Eq. (1) cites for example Ref. 46, where some of the terms were introduced. However, in that paper, surface charge density Q was specifically described as “areal hole density” i.e. describing a positive charge, whereas you seem to assume a negative charge. How can you apply the same equations to these two opposing cases? Why would in your case the negative charge be dominant, (which the authors in Ref 46 assume can be neutralized more easily)?*

Comment #8: *-In addition, you should describe already in the main text what each term in Eq (1) describes, and define every variable of the equations of the main text (including Eq (2)).*

Comment #9: *-Shouldn't the electrons of the flood gun be somehow part of the modelling of Eq. (1) ?*

Comment #10: *To understand Eq (2), the figure from Supplementary Note 3 is required in the main text.*

Responses to:

Comment #7

We thank the reviewer for these very important suggestions and questions that inspire us to have a thorough reconsideration of the ion emission model. After carefully reconsideration and further experiments, we believe that the positive charges injected by HFIB induce the primary ion emission, and the equation 1 cited from Ref. [46] is still employed to explain the positive surface charge density. Several reasons change our mind:

1. The charge carrier lifetime in amorphous SiO₂ is of 10 ns magnitude (main text Ref. [50, 51]), which means the electrons only survive for 10 ns before recombination with holes and ions. This makes the reabsorb secondary electrons impossible to accumulate and induce ion emission.
2. The secondary electrons (SEs) are reabsorbed by positive surface charge and gather near the surface when the SiO₂ layer thickness reaches 300 nm. When silicon dioxide thickness is down to 100 nm, no significant SE reabsorption takes place (main text Ref. [48]). This is consistent with our observation of Si wafer with 100 nm thermal oxidized (TOX) SiO₂ layer (Fig. R1a). The HIM images come from a SE detector with positive bias voltage to

collect the SEs excited by HFIB. Therefore, if reabsorption of SE takes place, the HIM image of the SiO₂ would be black. In Fig. R1(a) the SiO₂ substrate is brighter than IL droplet which is contrary to the results observed on the 300 nm or pure SiO₂ substrate employed in the manuscript (as shown in main text Fig. 1e). The results confirm that SE reabsorption does not happen on 100 nm SiO₂ substrate. As shown in Fig. R1, we successfully conducted IBFW experiments to induce IL film from liquid reservoir on 100 nm SiO₂ layer oxidized on Si wafer. Since no SE reabsorption happens on this substrate, we believe that not the negative charges but the positive charges account for the ion emission process involved in IBFW.

Fig. R1: HIM images of IL droplets on 100 nm SiO₂ and the optical image of the same droplets and film

Fig. 1e: The HIM image of IL pattern (brighter part) on PECVD SiO₂ (the black back ground, the reabsorption of SEs is so strong that the SiO₂ substrate can barely be observed) is quoted from main text to compare with Fig. R1.

3. According to main text Ref. [48], the electron beam also induces positive surface charges in SiO₂. This is due to the Secondary Electron Yield (which means the average number of SEs excited by one incident electron) of electron beam is larger than one, so with one electron injected to the sample more than one electrons are excited out from the sample.

Therefore, the overall charging effect of electron beam is positive. The electron beam can also induce the flow of ionic liquids (Supplementary Fig. 8), but the film pattern induced by electron beam is out of control. The diffractive nature of electron lead to a much wider surface charge lateral distribution may give rise to such results.

Based on these results, we have reconstructed the working principal section in our manuscript. The positive surface charges injected by HIM induce the primary ion emission and the IBFW phenomenon. We have also reconducted the MD simulation, using only positive charges injected and removed from the substrate to induce an identical film of ions and the movement of the droplet mass center. The equation we adopted from literature (old [46], now Ref. [54]) that modeling the positive charge of silica by HFIB irradiation is still adopted in our manuscript.

Comment #8 -*In addition, you should describe already in the main text what each term in Eq (1) describes, and define every variable of the equations of the main text (including Eq (2)).*

We expanded the working principal section so that every term of the equations in main text have been explained.

Comment #9 -*Shouldn't the electrons of the flood gun be somehow part of the modelling of Eq. (1) ?*

There are two reasons we did not include the flood gun current into calculation:

1. The flood gun can only work after the HFIB finishes an entire row (or entire frame) of scan spots and is always turned off during the HFIB irradiation, with a small beam current 0.5 pA. The NPVE scan spots array usually composes 1024×1024 scan spots, so the injection dosage of HFIB exceeds the flood gun at least three to six orders of magnitude.
2. The ion emission model considers HFIB scans a single spot or a very small area near the contact line to induce ion emission, the flood gun most likely is not working under such conditions.

Comment #10: *To understand Eq (2), the figure from Supplementary Note 3 is required in the main text.*

We have added the figure to the main text, and labeled as Fig. 2d.

“Working Principles of IBFW

When HFIB irradiates at or near the CTL of droplet, the Helium ions generate special charges distribution⁴⁸ in the SiO₂ substrate, the **positive surface charges** induce the primary ion emission from the IL reservoir (Fig.2a). **HFIB exhibits two distinguishable features compared with electron beam and other ion beams⁴⁸**. Firstly, the divergence of HFIB interacting with solid samples is much smaller than electron beam due to its larger mass⁴⁹, which lead to a more localized surface charging area and consequently a programmable control over liquid flow (Supplementary Fig. 8). Secondly, HFIB tends to penetrate sample and induce less damaging compared with Gallium beams⁴⁸. The penetration depth exerts significant influence on IBFW's application potential in electrochemistry field. For example, 30 keV He beams with hundreds nm stopping range can easily penetrate a 10 nm Au electrode deposited on SiO₂ wafer and induce patterned liquid flow without devastating effect to Au surface (Supplementary Fig. 9), while 30 keV Ga with stopping range less than 20 nm⁴⁸ is hard to penetrate metal films and can

easily cause damage to the electrode. During the HFIB irradiation, He ions interact with solid atoms and excite holes-electrons in the sample, then the He ions lose kinetic energy and rest within the stopping range. **Since the excited electrons in amorphous SiO₂ only survive 10 ns or less before the recombination takes place^{50,51}, the positive charges dominate the surface charging and account for the primary anion emission. Monte Carlo simulation results⁴⁸ reveal that the lateral projection length of 30 kV He ions irradiated on SiO₂ (also SiO₂ with 10 nm Au and 5 nm Ti layers) ranges from 90 nm to 110 nm (Supplementary Fig. 9a,b). The consequent positive charges (ions, holes) distribute over 10² nm in lateral direction and determine the ion emission and IBFW spatial resolution.**

When the HFIB ceases to irradiate, the surface charges dissipate due to the drainage current and the electron-hole recombination (Fig. 2b) then the **emitted anions** induce the **secondary cation emission**. Both ions meet ahead of the CTL and form an ultra-thin precursor film with thickness comparable to ion size at the irradiated area. As a matter of fact, when the electric field of surface charge is strong enough, not only separate ions but clusters or even tiny droplets contain both ions are emitted⁵² to scanned area and make up the precursor films. In both cases, the ultralow thickness of precursor film gives rise to the high disjoining pressure (10⁵⁻⁶ Pa) that irrigates and thickens the precursor film until be balanced by the capillary force (10²⁻³ Pa) and a continuous liquid film is formed (Fig. 2c).

The surface charging process of dielectric materials (SiO₂ for example) under the irradiation of focus ion beams (Ga⁺ or He⁺) was thoroughly discussed in literatures⁵³⁻⁵⁵, and the charging accumulation and dissipation is manipulated by the following factors: (1) generation of electron-hole pairs in the solid by incident ions; (2) neutralization of the incident ions by the excited free electrons; (3) sputtering of the surface atoms; (4) charging due to the secondary ion-electron emission; (5) leakage of mobile electron-hole pairs to the silicon substrate; (6) induced shallow traps by the incident ions and a consequent preferred trapping relative to the deep traps. The surface charge density (SCD) of SiO₂ at HFIB irradiation⁵⁴ can be expressed as a function of time (Supplementary Note 2):

$$\frac{dQ(t)}{dt} = P(1 + \gamma_e) \cdot I(t) - k \frac{Q(t)}{\epsilon_r \epsilon_0} - \frac{7}{4} Y I(t) \cdot \Omega_0 \frac{Q(t)}{R_p} - \int_0^t J(t) dt$$

(1)

The RHS composes of 4 terms. The first is the ion incident term which represents the electron-hole pairs accumulation induced by the ion incident and secondary emission, where P is the probability factor accounts for the electron-hole recombination, γ_e is the secondary electron emission yield of SiO₂, and $I(t)$ is the beam current density of HFIB. The second term is the leakage current term, where k is the conductivity of the substrate, ϵ_r is the substrate relative permittivity, and ϵ_0 is the vacuum permittivity constant. The third term is the sputtering yield induced charge reduction, where Y is the sputtering yield acquired from SRIM simulation, Ω_0 is the atomic volume which can be estimated by the average density of SiO₂, R_p is the ions stopping range from SRIM. The last term accounts for the accumulation of emitted counterions, where $J(t)$ is the ion emission rate at current SCD, which is often described as a kinetic process in which ions evaporate from liquid-vacuum interface. The emission current density reads⁵⁶:

$$j_e = \frac{k_B T}{h} \sigma \exp\left(-\frac{\Delta G - G(E_n^v)}{k_B T}\right), \quad (2)$$

where j_e is the current emitted per unit IL-vacuum surface area, k_B is Boltzmann's constant, T is the liquid temperature, h is Planck's constant, σ is the local net charge density at the liquid-vacuum interface, ΔG is the Gibbs free energy barrier for an ion to be emitted, E_n^v is the local vacuum electric field normal to the interface. $G(E_n^v)$ is the reduction of solvation energy barrier due to the external electric field, assumed to take the form $G(E_n^v) = \sqrt{\frac{q^3 E_n^v}{4\pi\epsilon_0} \frac{\epsilon_r - 1}{\epsilon_r + 1}}$ by the Schottky hump, where q is the ion's charge. The solvation energy of emitted ion can be estimated by the Born model as $\Delta G = \left(\frac{27}{4}\pi\right)^{1/3} \frac{\gamma^{1/3} q^{4/3} (1-\epsilon_r)^{2/3}}{(4\pi\epsilon_0)^{2/3}}$, where γ is the liquid – vacuum surface tension. Adopting the mechanical model proposed in the following paragraph, the electric field of SCD, E_n^v , can help to calculate the ion emission rate and the SCD.

A schematic diagram is shown in Fig. 2d to model the mechanical balance between IL surface tension and the electrostatic force exerted by the surface charge. The IL–vacuum surface is distorted by the electric field and forms a bumping meniscus. When the meniscus is distorted to be hemispherical, the vertical component of surface tension reaches maximum. Once the surface charge continues to increase, a significant ion emission would take place during which both ions, clusters and tiny droplets may emit from the interface^{52,56,57}. The distance between the scan spot (center of surface charge) and the contact line ranges from 10^0 to 10^2 nm, as long as the surface charge is strong enough to induce ion emission. The critical SCD that can induce significant ion emission depends on the distance between the scan spot and the reservoir CTL (more detailed deduction can be found in Supplementary Note 3):

$$Q_{surf} \cong \frac{(d+r^*)^2 + l_0(d+r^*)}{k_0 \cos^3 \alpha} E^* \quad (3)$$

Where d is the distance between the scan spot and the CTL; $r^* = \frac{q^6 \gamma}{4\pi^2 \epsilon_0^3 (\Delta G)^4} \sim 10^{-8} m$ is the characteristic ion emission radius derived in literature⁵²; l_0 is the surface charging area length scale, which represents the lateral distribution of positive surface charges⁴⁸; k_0 is the Coulomb constant; α is the ion emission angle as depicted in Fig. 2d; and $E^* \sim 10^9 \sim 10^{11} V/m$ is the characteristic electric field⁵⁶ for significant ion emission. At given separation distance d , the critical SCD can be calculated by equation 3. By invoking equations 1 and 2, the dosage of HFIB required for the critical SCD can be calculated.”

(Page 4-6, Main text)

Figure 2: Working principle of IBFW nano-printing. (a-c) Schematic of the IBFW working principle. (a) When HFIB irradiates the CTL, the **positive surface charging induces the primary anion emission**. (b) When HFIB ceases to scan, surface charging dissipates and the emitted anions induce the **secondary cation emission**. (c) The emitted ions from previous stages form an ultra-thin precursor film, and the consequent disjoining pressure propels and stabilizes liquid film. (d) The **mechanical balance between the surface charge density induced electrostatic force and the surface tension of IL-vacuum interface**. The distance between HFIB scans spot and contact line is d ; the surface charging uniformly distributes over a region with length scale, l_0 ; the surface-charge-exert electrostatic force and distorts the IL-vacuum interface, balanced by the surface γ ; α is the angle between substrate the connection line of SCD center and ion emission center; r^* is the ion emission critical radius, and is also the vertical distance between ion emission center and substrate. (e) Experimental HFIB dosage density to induce IL flow (ions per square of nm) and calculation results of critical dose density to induce significant ion emission. The inset shows when the scan position is separated too far from the CTL, the consecutive liquid film degenerates to the local protrusion flow as shown in Fig. 1b. (f) IBFW induced liquid film lengths (μm) on substrates with different conductivities. The right ordinate represents the overall ion emission number calculated by the beam parameters and sample characters. (g) Molecular dynamics simulation of [EMIM][DCA] droplet (640 ion pairs) deposited on fused silica substrate going through surface charges injection and removal. The arrows indicate the most directed movements of ions: the pale blue arrows at the beginning stage represent the surface charge induced **primary anion emission**; the purple arrows of **cations** represent the **emitted anions induced secondary cation emission**. The shaded regions (red)

represent the **surface injection region with positive charges**. (h) IBFW film flow speed as a function of liquid film length. The single spot experiments are conducted with a line pattern of scan spots, at constant beam current 1 pA, dwell time 10 μ s, spacing 1 nm. The change width experiments are conducted with rectangular pattern of scan spots with constant length 20 μ m and different width, while keeping the beam current beam current 1 pA, dwell time 2 μ s, spacing 1 nm constant to keep the dose density unchanged. The blue shaded region is the range between the calculate velocity lower and upper limit.

(Page 22-23, Main text)

“The influence of substrate conductivity over ion emission and IBFW film inducing is clear.

There are two reasons we did not include the flood gun current into calculation:

1. The flood gun can only work after the HFIB finishes an entire row (or entire frame) of scan spots. The NPVE scan spots array usually composes 1024 \times 1024 scan spots, so the injection dosage of HFIB exceeds the flood gun three to six orders of magnitude.

2. The ion emission model considers HFIB scans a single spot or a very small area near the contact lien to induce ion emission, the flood gun most likely is not working under such conditions.”

(Page S7, Supplementary)

Comment #11: *How did you measure the experimental surface charge density of Fig. 2 D ? Could you experimentally determine the sign of the charge? If you used Eq (1), how is that an “experimental ” result?*

Response: We thank the reviewer for the questions and comments. The experimental surface charge density (SCD) in Fig. 2d was the critical HFIB injection dose density that takes to induce consecutive liquid film from the reservoir. We corrected this mistake, and recalculated the critical HFIB dose density to induce significant ion emission to compare with our experimental result. The experiments were conducted with beam current 1 pA and scan spacing 1 nm, and the dwell time was used to adjust the irradiation dose density which can be directly calculated: $D_{experiment} = I \cdot \tau \cdot s^{-2}$. Where I is beam current, τ is dwell time and s is scan spacing. To compare with the experimental dose density, we employ equation 3, $Q_{surf} \cong$

$$\frac{(d+r^*)^2+l_0(d+r^*)}{k_0 \cos^3 \alpha} E^*,$$

to calculate the critical surface charge density at different distance d between scan spot and contact line. Invoking equation 1, $\frac{dQ(t)}{dt} = P(1 + \gamma_e) \cdot I(t) - \sigma \frac{Q(t)}{\epsilon_r \epsilon_0} -$

$$\frac{7}{4} Y I(t) \cdot \Omega_0 \frac{Q(t)}{R_p} - \int_0^t J(t) dt,$$

we can calculate the dwell time and the dose density it takes to reach the critical surface charge densities at given distance d . We corrected the words and terms we used in the main text and changed Fig. 2f to compare the calculated dose density with the experiment dose density.

The measurement of sign of charge within the HIM chamber with the Helium ion source powering on is risky. We demonstrated experimentally in Figure R1 that the reabsorbed SE is not the driving force of IBFW, and positive charges of dielectric materials at the irradiation of Helium ion beam is well accepted. Therefore, we did not conduct further experiment to test the

sign of charge.

“At given separation distance d , the critical SCD can be calculated by equation 3. By invoking equations 1 and 2, the dosage of HFIB required for the critical SCD can be acquired.

The relationship between critical HFIB dose density to induce IBFW, and the distance between starting scan spot and droplet CTL, d , is calculated employing equations 1 to 3 with the results depicted in Fig. 2e by blue line. The beam current employed in calculation is 1 pA, scan spacing is 1 nm, and the dwell time determines the calculated dose density. Here, we assume that the critical dose density that takes to induce significant ion emission coincides with the dose to achieve IBFW. The experiments are conducted to measure the critical HFIB dose density with 1 pA beam current and 1 nm spacing to induce continuous liquid film from reservoir with the results shown in Fig. 2e by orange rhombuses. The calculation results agree with the experimental critical dose density qualitatively, and confirm our hypothesis that the surface charging induced ion emission accounts for the IBFW.”

(Page 6, Main text)

Figure 2 (e) Experimental HFIB dosage density to induce IL flow (ions/nm²) and calculation results of critical dose density to induce significant ion emission. The inset shows when the scan position is separated too far from the CTL, the consecutive liquid film degenerates to the local protrusion flow as shown in Fig. 1b.

(Page 22, Main text)

Comment #12: “The qualitative consistency ... verifies the mechanism we propose.” Big statement, considering that you refer to Fig. 2E where the agreement is NOT good. (i.e. modelling doesn't predict at all differences between different insulating substrates)

Response: We thank the reviewer for the comments. The ‘qualitative consistency’ just means

that our crude model can distinguish insulating, semi conductive and conducting substrates. The ion emission model we proposed fails to explain the differences between different insulating substrates due to other factors that have not been considered in our model. For example, the solid surface roughness elements distort the contact line and the IL-vacuum interface. The distorted interface with different curvature can influence the ion emission energy barrier (main text Ref. [68]). Therefore, we discovered in experiments that at some positions of CTL, film can be easily produced, while some other positions of the same droplet cannot. This is one of the possible reasons that lead to the different behavior of different insulating material substrates. We changed the words we used in the main text, and explained the shortage of our model in the Supplementary Note 3. Currently, our simplified models cannot distinguish different insulating materials, where some factors may have not included.

“The orange rectangles represent the lengths of liquid film that can be induced on different substrates under identical HFIB treatment. IL film cannot be induced on pure conducting samples such as Au and Cu, liquid extends slightly on semiconductor (Si), whereas liquid film propagates a long distance on insulated substrates *e.g.*, quartz and mica. Employing equations 1-3, we can calculate the numbers of emitted ions during a single spot scan of HFIB (Supplementary Note 2, Supplementary Fig. 10), and is depicted in Fig. 2f as blue circles. As can be seen, **the ion emission model we propose herein can qualitatively explain the different effects of IBFW on conducting, semi-conducting and insulating substrates. Yet the IBFW effect is also influenced by other substrate situations (roughness, for example), and the differences between various insulated substrates are failed to be captured by the simplified 2D model.**”

(Page 6, Main text)

“The solvation energy of emitted ion can be estimated by the Born model as $\Delta G = \left(\frac{27}{4}\pi\right)^{1/3} \frac{\gamma^{1/3} q^{4/3} (1-\epsilon_r)^{2/3}}{(4\pi\epsilon_0)^{2/3}}$, where γ is the liquid – vacuum surface tension. **Noteworthy, the ion emission energy barrier can be altered significantly by the local curvature of the IL-vacuum surface²⁵, but we consider a simple 2-dimensional model in current work and exclude this factor. Such influence should be considered to explain the different performances of IBFW at the CTL of the same liquid reservoir in the future.**”

(Page S9-10, Supplementary)

Comment #13: *“The relationship between flow velocity and film length can also be measured experimentally ” . I don ’ t understand how you can measure the flow velocity in your experimental setting.*

Response: We thank the reviewer for the important comments. Two methods were adopted to measure the flow velocity, and are illustrated here:

The HFIB scans the designed pattern row by row, so the beam speed vertical to the CTL can be calculated as:

$$v_{beam} = \frac{s_{vertical}}{N_{row} \cdot \tau + V_{refresh}}, \quad (5)$$

Where, $s_{vertical}$ is the vertical scan spacing, N_{row} is the number of scan spots in one row, τ

is the dwell time that HFIB stay at a single spot, $V_{refresh}$ is a small time (10 μ s) that software takes to reset the HFIB ready for the next row of scan. If the vertical speed of HFIB exceeds the film velocity, the distance between the scan spot and the liquid film CTL would increase until the scan spot is too far ahead of the film and the liquid would cease to flow. The critical interaction distance with given beam parameters can be determined experimentally (Supplementary Fig. 14a). Due to the pronounced impact of dose density on the flow velocity of liquid film (Supplementary Fig. 14b), the flow velocity measurements are conducted under the same dose density by keeping the beam current $I = 1$ pA, scan spot spacing $s = 1$ nm and dwell time $\tau = 2$ μ s constant and only alter the width of the rectangle pattern. The pattern width controls the N_{row} and consequently alters the beam vertical speed. By scanning rectangle patterns with same length but different widths outwards from the reservoir CTL, the vertical speed of beam can be changed at constant HFIB dose density. The IBFW film length decreases with the beam speed increasing (Supplementary Fig. 14c), and consequently the average flow velocity of films with different length can be measured. The change width measurement results are shown in Fig. 2h by the hollow orange stars. The results of film speed at extremely long film length are acquired by first fabricating a long liquid film (300 μ m \times 10 μ m, 600 μ m \times 10 μ m, and 900 μ m \times 10 μ m respectively) from the reservoir, then the change width measurements are conducted at the front of the long film. Since the fabrication of extreme long film can be time consuming, these data are only measured once. The other experiments are repeated for at least ten times with the average value and standard error shown in Fig. 2h.

The shortage of the change-width method is that the NPVE scan pattern assembling limits the maximum velocity the beam can move vertically. To overcome such limitation, we adopt single spot scan method. In which, a line pattern made up by a series of scan spots is used. The scan speed is altered by changing the vertical refresh time between each scan spot, while keeping beam current 1 pA, dwell time 10 μ s and spacing 1 nm all constant. The dwell time is elongated to compensate the dose density reduction, since the scan area is influenced by neighboring scan spots in a rectangular pattern. All single spot measurements are repeated at least five times. The results of single spot scan are represented by the orange stars in Fig. 2h.

Apart from velocity measurement explanation, the flow model is modified by the introduction of boundary slip length of IL-SiO₂ interface according to other Reviewer's comment. Since the slip length vary from 2 nm to over 16 nm determined by the combined surface conditions (Ref. [58-60]), the possible flow velocity range is calculated employing the lower and upper limit of IL slip length as shown by the blue region in Fig. 2h.

“We next verify that the disjoining pressure propels and stabilizes the nanofilm. The propagation speed of IBFW liquid film decreases monotonically with the increase of film length:

$$U \sim \frac{h^2 + 3bh}{3\mu} \cdot \frac{\Pi(h_{min}) - \gamma\kappa}{L} \quad (4)$$

Where U is the average flow speed, h is equilibrium film thickness, b is the slip length of IL-SiO₂ interface⁵⁸⁻⁶⁰, μ is IL viscosity, $\Pi(h_{min})$ is disjoining pressure at minimum film thickness h_{min} , L is film length, γ is IL surface tension, and κ is curvature of IL-vacuum interface at the

conjunction of film and reservoir (Supplementary Table. 2, Supplementary Note 4, Supplementary Fig. 13). We depict the calculation results in Fig. 2h. Since the boundary slip length of IL-SiO₂ interface depends on the combined surface conditions and ranges from 2 nm to over 16 nm, we employ the lower (2 nm) and upper (16 nm) limits of slip length to give an estimation on the possible range of flow speed in Fig. 2h as the blue shaded region, and the results of average value 10 nm⁵⁸ is shown by the deep blue line.

The relationship between flow velocity and film length can also be measured experimentally. As shown in Supplementary Fig. 4, the HFIB scans the designed pattern row by row, so the beam speed vertical to the CTL can be calculated as:

$$v_{beam} = \frac{s_{vertical}}{N_{row} \cdot \tau + V_{refresh}}, \quad (5)$$

Where, $s_{vertical}$ is the vertical scan spacing, N_{row} is the number of scan spots in one row, τ is the dwell time that HFIB stay at a single spot, $V_{refresh}$ is a small time (10 μ s) that NPVE takes to reset the HFIB for next row of scan. If the vertical speed of HFIB exceeds the film velocity, the distance between the scan spot and the liquid film CTL would increase until the scan spot is too far ahead of the film and the liquid would cease to flow. The critical interaction distance with given beam parameters can be determined experimentally (Supplementary Fig. 14a). Due to the pronounced impact of dose density on the flow velocity of liquid film (Supplementary Fig. 14b), the flow velocity measurements are conducted under the same dose density by keeping the beam current $I = 1$ pA, scan spot spacing $s = 1$ nm and dwell time $\tau = 2$ μ s constant and only alter the width of the rectangle pattern. The pattern width controls the N_{row} and consequently alters the beam vertical speed. By scanning rectangle patterns with same length but different widths outwards from the reservoir CTL, the vertical speed of beam can be changed at constant HFIB dose density. The IBFW film length decreases with the beam speed increasing (Supplementary Fig. 14c), and consequently the average flow velocity of films with different length can be measured. The change width measurement results are shown in Fig. 2h by the hollow orange stars. The results of film speed at extremely long film lengths are acquired by first fabricating a long liquid film (300 μ m \times 10 μ m, 600 μ m \times 10 μ m, and 900 μ m \times 10 μ m respectively) from the reservoir, then the change width measurements are conducted at the front of the long film. Since the fabrication of extreme long film can be time consuming, these data are only measured once. The other experiments are repeated for at least ten times with the average value and standard error shown in Fig. 2h.

The shortage of the change-width method is that the NPVE scan pattern assembling limits the maximum velocity the beam can move vertically. To overcome such limitation, we adopt single spot scan method. In which, a line pattern made up by a series of scan spots is used. The scan speed is altered by changing the vertical refresh time between each scan spot, while keeping beam current 1 pA, dwell time 10 μ s and spacing 1 nm all constant. The dwell time is elongated to compensate the dose density reduction, since the scan area is influenced by neighboring scan spots in a rectangular pattern. All single spot measurements are repeated at least five times. The results of single spot scan are represented by the orange stars in Fig. 2h. The consistency between the calculation and experiments suggests that disjoining pressure can explain the propagation of IBFW film.”

(Page 7-8, Main text)

“Figure 2: (h) IBFW film flow speed as a function of liquid film length. The single spot experiments are conducted with a line pattern of scan spots, at constant beam current 1 pA, dwell time 10 μs , spacing 1 nm. The change width experiments are conducted with rectangular pattern of scan spots with constant length 20 μm and different width, while keeping the beam current beam current 1 pA, dwell time 2 μs , spacing 1 nm constant to keep the dose density unchanged. The blue shaded region is the range between the calculate velocity lower and upper limit due to the range of slip lengths^{58,59}, the deep blue line is the calculation result employing the average slip length⁵⁸.”

(Page 23-24, Main text)

“Supplementary Figure 14: Flow speed measurement experiments.

(a) The single-spot scan style in HIM, which scans one-dimensionally along the pattern direction, is applied to analyze the interaction spatial range of a single irradiation spot quantitatively. The critical maximum spacing between neighbor scan spots to induce liquid

flow, s_c , represents the upper limit for the interaction spatial range of the beam spot. When scan spot spacing exceed s_c , HFIB fails to induce continuous flow however large the dose is. The injection dose density, D , is regulated by changing beam current, I , at constant dwell time 100 μs , or changing dwell time, τ , at constant beam current 0.7 pA. The relationship between s_c and D of each beam spot is plotted. (b) Maximal flow velocity as a function of the beam current I for a 20 μm length liquid film. The scan speed is changed by the pattern width. The flow speed increases with increasing beam current at same dwell time 1 μs . (c) **An example of the change width method for the measurements of flow velocity at different film length.**

(Page S31, Supplementary)

Comment #14: *AFM measurements “The liquid-vacuum interface ...” Is the AFM measurement done in vacuum instead of air?*

Response: We thank the reviewer for the question. The AFM measurements were conducted all in atmosphere condition. We have corrected the mistake in the main text.

“We employ atomic force microscope (AFM) to manifest the nanoscale flow control of IBFW. Fig. 3a shows the front 7 μm of a 28 $\mu\text{m} \times 1 \mu\text{m}$ IL film with thickness around 30~40 nm (an average of 35.4 ± 1.7 nm). The film thickness remains unchanged along the flow path (Fig. 3A). The film width is 1 μm , coincides with designed pattern. The **liquid-air interface** is much smoother than solid substrate (RMS roughness 7.9 ± 5.7 nm). The minimal line width of IBFW film reaches 106 nm (Supplementary Fig. 15), and is limited by the surface charges spatial distribution which was reported around 10^2 nm¹⁵. If the surface charges can be trapped within a narrower spatial range, the ideal line width limitation may be comparable with film thickness.”

(Page 8, Main text)

Comment #15: *“The current in nanofilm linearly depends on the humidity”. What is the mechanism?? Anyways, it seems the current is not only a function of humidity as it depends directly on time as well based on Fig. S18. How could you disentangle the effect of time from the effect of humidity in a real sensor as some voltage bias history dependence seems to exist (referring to Fig. S18F)*

Response: We thank the reviewer for the important questions. We believe the reaction-diffusion competition differences and the higher adsorption rate endowed by higher surface-volume ratio of nanocircuit leads to the differences in the humidity response curve we measured. For the droplet circuit, the long distance for the water molecules to diffuse to the electrode restricts the reaction rate, therefore, the change of current with humidity is not obvious. Nevertheless, the thin liquid film circuit with much shorter diffusion distance largely accelerate the diffusion process, which is the rate-determining step of current experiment. Moreover, the larger surface-volume ratio benefits the adsorption of the vapor molecules and further improve

the reaction current. We have expanded the paragraph in main text to further illustrate the mechanism.

The time dependent current measured in Supplementary Fig. S18 is due to the gathering of counterions to the electrodes to form an electric double layer (EDL) when the voltage is switched on. If the voltage has been switched on for a long time, the EDL at the electrodes-IL interface comes into a thermodynamics equilibrium would electrostatically block the electrode voltage, and no measurable current can be detected. Employing a sensing circuit that has reached equilibrium (with bias voltage turned on for a long time, 1h is more than adequate) will disentangle the time effect.

The bias history dependence comes from repeatedly switched on and off the voltage, which is another mistake we made during the caption of Supplementary Fig. 18. When bias voltage switched on, counterions gathering to form an EDL and give rise to a current jump. As the EDL developing, the electrode voltage is gradually being blocked and the current decay to near zero until reaches equilibrium. When voltage switched off, the EDL will gradually disappear and return to the initial condition. If the rest time is long enough, the history dependence would disappear. But the rest time in our experiment was too short, and the EDL has not reached equilibrium before the bias voltage is exerted again. When the bias voltage turned on again, the real voltage feels by the IL is actually smaller than the bias voltage we exert since part of the old EDL still absorbed at the electrode surface, which leads to a history dependence in Supplementary Fig. 18f.

“In Fig. 4a, b and Supplementary Fig. 19, we present a room-humidity-sensing circuit utilizing IBFW technique. The water molecules are adsorbed by the IL then diffuse to and react at the electrode surfaces, generating a reaction current I_{ds} . The source-drain currents of a IBFW nanofilm circuit and a micrometer-size droplet circuit are measured within the same chamber with the relative humidity ranging from 40% to 70%. The current in nanofilm linearly depends on the humidity, while no significant change can be observed for the micrometer droplet. **The competition between the adsorption at liquid-air interface and the diffusion within liquid circuit can explain such differences.** For the droplet circuit, the long distance for the water molecules to diffuse to the electrode surface restricts the reaction rate, therefore, the change of current with humidity is not obvious. Nevertheless, the thin liquid film circuit with much shorter diffusion distance and relaxation time largely accelerates the diffusion process, which is the rate-determining step of the current experiment. **Moreover, the much higher surface-volume ratio benefits the adsorption of the vapor molecules and further improves the reaction current sensitivity.** Both the sensitivity and response speed are greatly enhanced due to the size effect endowed by IBFW nano film. The simple device presented here can manifest the feasibility of the IBFW circuit in sensing circuit manufacturing.”

(Page 9, Main text)

Comment #16: *Fig. 3F use real current units instead of dimensionless and tell what the bias voltage used was. BTW Fig. S18 C has wrong units (A vs nA). Fig. S18G has no caption, what is it?*

Response: We thank the reviewer for the suggestions. We corrected the units to dimensionless

in Fig. 4b, and the voltage exerted between the two Au electrodes was 1 V. We have corrected the wrong units used in Supplementary Fig. 19 a and b (used to be labeled as Fig. S18). The captions for the previous version of Supplementary Fig. S18 f and g were ambiguous and wrong. Fig. S18 f was the Drain current I-t curve with the source drain voltage, 10 V, switched on and off periodically (turned on 480s, then off 20s) for four times. Fig. S18 g was the Drain current I-t curve with constant source-drain voltage 10 V. Supplementary Fig. 19 has been reconstructed.

We think part of the old version was irrelevant to illustrate the difference between nano circuit and micro droplet. In current version, Supplementary Fig. 19c is the drain current I-t curve of a droplet circuit and a nano-film circuit in atmosphere and in vacuum chamber at constant source-drain voltage 10 V. Both the droplet circuit and nanofilm circuit have a higher current in atmosphere than in vacuum chamber, this is due to the water molecules adsorbed by the droplet and the film that react at the electrode surface and create a reaction current. The result in vacuum chamber is more intriguing. The adsorption-reaction current has been disentangled, and the current we measured is solely contributed by the formation of electric double layer. As shown by the yellow curve, the EDL formation current curve of nanofilm circuit quickly decreases to be indiscernible, which indicate that the EDL in nanofilm circuit established quickly and reaches an equilibrium. On the contrary, the EDL formation current of droplet circuit decrease much slower, and keep on decreasing at the end of the 300 s measurement, which indicate that the equilibrium is hard to attain in a droplet circuit. Such differences explain that nano film circuit with a much shorter relaxation time is more suitable for gas sensing purpose. Supplementary Fig. 19d is an HIM image of the nanocircuit fabricated.

Fig. 4 (a,b) Proof-of-concept gas sensing prototype. (a) the IBFW liquid nanofilm connected two droplets of IL settled on two Au electrodes for electrochemistry data collection. (b) Results of IBFW liquid nano-circuit in gas sensing and compared with a micro meter droplet settled between and connects two identical Au electrodes. The inset shows the schematic of a room-humidity sensor based on IBFW fabricated nanofilm. The drainage-source currents of both microdroplet and nanofilm are illustrated and the linear fitting results are depicted as dash lines. **The voltage exerted between two electrodes is 1 V.** The background color map represents the relative humidity of test chamber ranges from 40% to 70%.

(Page 26-27, Main text)

Supplementary Figure 19: Volt-Ampere characteristics analysis of IL nano film circuit.

(a) Drain current between Au electrodes vs. time, the black line represents an identical measurement with no liquid film connect the electrodes; (b) linear sweep volt-ampere curve of liquid film circuit; (c) The transient current curves of IL film circuit and droplet circuit in high vacuum and in atmospheric environment as a function of time, the source-drain voltage is 10 V. Both the droplet circuit and nanofilm circuit have a higher current in atmosphere than in vacuum chamber, this is due to the water molecules adsorbed by the droplet and the film that react at the electrode surface and create a reaction current. The result in vacuum chamber is more intriguing. The adsorption-reaction current has been disentangled, and the current we measured is solely contributed by the formation of electric double layer. As shown by the yellow curve, the EDL formation current curve of nanofilm circuit quickly decreases to be indiscernible, which indicate that the EDL in nanofilm circuit established quickly and reaches an equilibrium. On the contrary, the EDL formation current of droplet circuit decrease much slower, and keep on decreasing at the end of the 300 s measurement, which indicate that the equilibrium is hard to attain in a droplet circuit. Such differences explain that nano film circuit with a much shorter relaxation time is more suitable for gas sensing purpose. (d) The HIM image of the liquid nano circuit employed in this experiment.

(Page S36, Supplementary)

Responses to Reviewer #3:

The authors developed a new method to perform rewritable surface printing of ionic liquids. The method relies on the disjoining pressure to induce surface films of nanometer thickness, and utilizes helium ion beams to initiate the flow of liquids and pattern creation. The manuscript comprehensively characterizes and explains the underlying mechanism of this printing process. However, as an application-driven work, the prospect of practical real-world applications is a bit weak. Although the authors demonstrated a sensing device application (Fig. 3F), it only utilizes the nanoscale thickness of the ionic liquid film, and the surface patterning is not very relevant. Specific concerns include:

Response: We thank the reviewer for the positive comments on our manuscript.

Comment #1: *This method utilizes ionic liquids, and seems hard to extend to other liquid systems due to the high-vacuum requirements of the helium ion microscope. Ionic liquid as of now are still quite expensive, limiting the range of applications. Are there ways to improve or revise the method so that other liquid or even solid structures can also be printed?*

Response: We thank the reviewer for these inspiring suggestions. The liquid working media can be expanded by injection of aqueous or organic solution into the liquid reservoir of the IBFW liquid film. The film pattern can act as a stable flow channel for the solutions injected, we fabricated the in-situ chemical reaction chip to serve as a proof-of-concept prototype. Moreover, the electric field induced Taylor cone and ion emission are ubiquitous for many liquids including water. If an appropriate capsule can be fabricated with windows transparent to ion beams, we believe many other liquids can be employed in IBFW experiments.

The IBFW liquid film pattern can be electrochemically deposited to be solid particles with designed patterns. We added the ‘Applications of IBFW’ in the main text to discuss in details. The ILs is more expensive than water or ethanol, but many types of them are still cheap enough for research. The [EMIM][DCA] we used in our experiments were purchased at a price of 24 CNY ~ 3 USD for 20 g.

We listed the modifications we made to our manuscript that can answer the reviewer’s questions.

1. We modified the introduction to introduce potential application fields suitable for ionic liquids.
2. We added a new section ‘Applications of IBFW’ for further demonstration. We used IBFW to fabricate liquid channels as a guidance for aqueous solutions of reagent which were later injected into the IL droplets for the in-situ colorimetric reactions of SCN^- with various metal ions. This experiment demonstrates that IBFW liquid film pattern can act as a stable flow channel for later injection of analyte solutions.
3. We conducted electrochemical experiments to reveal that IBFW is also capable of transforming liquid film pattern into desired solid materials. Electrochemistry analytes are dissolved in 1-ethyl-3-methylimidazolium bis(trifluoromethylsulfonyl)imide ([EMIM][NTf₂]). The solution droplet was settled on ultra-thin metal film (15 nm)

deposited on SiO₂ wafer, and film patterns were printed by IBFW. The following electrochemical deposition transforms the liquid pattern into solid materials. Since the stopping range of 30 kV Helium ions exceeds 290 nm, a 15 nm metal film exerts no significant influence on the surface charge injection of HFIB, and consequently the IBFW can be conducted on very thin metal film. Monte Carlo simulation results were added to explain the penetrating capability of He ions (Supplementary Fig. 9b). We also discussed the potential influence of electrowetting on the IBFW film, and drew the conclusion that the voltage we adopted won't overcome the contact angle hysteresis and the IBFW films were not influenced by the deposition voltage.

For now, we only tested Ag nano-particles and AgTCNQ (7,7,8,8-tetracyanoquinodimethane) complex, but more electrochemistry tests can be undertaken to test the capability of solid deposition of both organic and inorganic materials.

“Based on the IBFW inducing mode, we develop a nano-printing technique of ILs, with film thickness down to 20~30 nm, minimal line width about 100 nm and corner radius down to 20 nm, and compare its performances with the reported methods. Besides, ILs are also known for their unique properties such as wide electrochemical potential window, high ionic conductivity, low toxicity and thermostability. These features make ILs increasingly important as electrolytes for lithium battery^{40,41} and electrodepositions of various materials ranging from metal nanoparticles^{42,43}, metal organic complexes⁴⁴ to conducting polymer films⁴⁵. We further demonstrate the IBFW as a versatile tool for various application fields including gas sensing circuit, in-situ chemical reaction chip, and electrochemical deposition of solid materials with desired patterns. The simplicity and versatility of IBFW technique suggests prospect in a range of liquid manipulation applications. By combining with electrochemical procedures, such technique can not only produce patterned liquid film but also solid materials which reveals possibility in nano-transistors fabrication⁴⁶, energy devices⁴⁰ and immunosensor circuit printing⁴⁷. We expect this technique can open a new avenue for applications in nano-printing and nano-circuit manufacturing.”

(Page 2-3, Main text)

“Applications of IBFW

As discussed previously, IL film pattern prepared by IBFW technology manifests three distinguishable features. First, the ultralow film thickness down to 30 nm indicates a high surface-volume ratio which is a key role in improving the gas sensing circuit sensitivity. Second, the capability of fabricating liquid film with desired pattern in a programmable and rewritable manner, which is important for in-situ chemical reaction and microfluidics chips. Third, the ILs are widely used in electrochemistry and reveal the possibility of transforming liquid film pattern into various solid materials ranges from organic to inorganic compounds.”

(Page 9, Main text)

“To demonstrate the potential of IBFW for microfluidics chip fabrication, we design an in-situ chemical reaction micro-chip. In Fig. 4c, the schematics shows a crosshair shaped micro fluid channel connects four separated droplets in four directions. Four square expansion windows are made on each part of the channel for the convenience of observation. The top-left inset

shows the chip with four droplets on finger-tip. After the IBFW fabrications, 0.1 μL of sodium thiocyanate solution (NaSCN 0.1 M in deionized water) is injected into the top droplet, and serves as the colorimetric reagent for the detection of and in-situ reaction with different metal ions. After the injection of NaSCN, 0.1 μL of 0.1 mM Fe^{3+} solution, 0.1 mM Cu^{2+} solution and 0.1mM Co^{2+} are injected into the bottom, left and right droplets respectively. The microchip is rested in atmosphere for 20 min for the metal ions fully diffuse into the channels and react with SCN^- within different square windows with the ion names printed previously. Then the sample is transferred into vacuum chamber for 48 h to diminish the water content in the solution system, which will alter the hydration status of the metal ion complexes and improve the colorimetric visibility. The red complex $\text{Fe}(\text{SCN})_3$ deposits in the bottom window. The gray deposition in the left window is complex $\text{Cu}(\text{SCN})_2$. And the blue deposition in the right window is complex $\text{Co}(\text{SCN})_2$. As shown in Fig. 4c, the SCN^- participates into 3 different reactions within several hundreds of micrometers flow channel. The time series pictures are shown in Supplementary Fig. 20, and the liquid film patterns remain unchanged during the experiments which last for over one month. The IBFW fluid channel exhibits great stabilities against vacuum/air transferring, the injection of solutions into droplet reservoir, and gravity. Such behavior demonstrates the robustness of the IBFW liquid film. **More importantly, all reagents are dissolved in deionized water then injected into the IL droplets and diffuse into the IL flow channel. This experiment demonstrates that IBFW liquid film pattern can act as a stable flow channel for later injection of analyte dissolved in water, ethanol and various molecular liquids due to the amphiphilicity of ILs. Such results greatly broaden the potential application fields for IBFW.”**

(Page 10, Main text)

Figure. 4: (c) In-situ chemical reaction chip. The left part is the schematics of the chip. Crosshair channels connect four droplets which are used to inject reagent water solutions. 0.1 μL of 0.1 M NaSCN, 0.1 mM Co^{2+} , 0.1 mM Fe^{3+} and 0.1mM Cu^{2+} are injected clockwise into four droplets (top, right, bottom, left). The inset on the top-left compares the size of microchip with fingertip, with red circle shows the four droplets. The right part is the optical image of the reaction chip that has been stored in vacuum chamber for 48 h after the injection, the color of complexes is more obvious with water removed from the system.

(Page 27-28, Main text)

Supplementary Figure 20: Time series images of in-situ chemical reaction chip.

(a) HIM image of channels right after the IBFW fabrication. (b) The fluid channel first transferred to the optical microscope; the colorless transparent fluid channel is a bit hard to be distinguished from the transparent silica substrate. (c) After the injection of all solutions and reacts in atmosphere for 5 min. (d) The whole picture of the reaction chip before transferred to vacuum chamber. (e) 48 h after metal ions injection and storing in vacuum. After stored in vacuum chamber, the water has been eliminated, and the color of the different complexes become obvious. The sample goes through several times of transferring between vacuum chamber and air environment and the injection of analyte solutions into droplet reservoirs, while the liquid film pattern remains barely changed through the experiments. Such results demonstrate the stability of IBFW liquid film channel to the exposure of air environment and to non-direct physical contact for solution injections. (f) The reaction chip one month after injection (one week for gravity stability test). The chip is mounted on a customized sample holder which can adjust the tilting angle from 0-180° to test the IBFW film stability against gravity. The inset shows the mounted sample with tilting angle 120° and is stored for one week.

(Page S37, Supplementary)

“Due to their unique properties, ILs have been proved to be an important category of solvent and electrolytes. Here we demonstrate that, by further combining with electrochemical procedure, the IBFW also manifests the capability of transforming liquid film pattern into various solid materials. Fig. 4d shows the schematics of a three-electrode electrochemistry experiment. A droplet of 1-ethyl-3-methylimidazolium bis(trifluoromethylsulfonyl)imide ([EMIM][NTf₂]) serves as the solvent of possible analytes, which are Silver bis(trifluoromethylsulfonyl)imide (Ag[NTf₂]) or a mixture of Ag⁺ and 7,7,8,8-tetracyanoquinodimethane (TCNQ) in our experiments. The liquid film pattern is fabricated by the IBFW technology onto the thin gold electrode and would be subsequently transformed into

nanoparticles. A 10 nm Au and 5 nm Ti film deposited onto SiO₂ serves as the working electrode and is connected to the workstation by a Pt probe. As revealed by the MC simulation results (Supplementary Fig. 9 b), the He ions with vertical stopping-range exceeds 290 nm can easily penetrate the 15 nm metal film and deposit positive charges into the 300 nm SiO₂ layer underneath. Therefore, the IBFW can be achieved on an ultra-thin metal film deposited on insulating substrate, and is not contradict with the conclusion that pure conducting substrates lead to the failure of IBFW shown in Fig. 2f. The counter electrode is a Pt probe emerged in IL, and a silver-plated probe serves as a pseudo reference electrode.

In Fig. 4e, we present an example of solid particles deposited from IBFW liquid film. The upper part shows silver nanoparticles with designed pattern (a 20 μm × 3 μm channel and PKU letters) potentiostatically deposited at -0.2 V (vs. Ag) for 180 s onto the gold electrode surface. And the lower part shows blue AgTCNQ particles on gold surface make up a 50 μm × 10 μm rectangular pattern deposited at -0.1 V (vs. Ag). **Noteworthy, the electrowetting phenomenon is ubiquitous in IL-electrode systems with contact angle hysteresis ranges from several to tens of degrees⁶³.** When the voltage applied between the IL and electrode surface is large enough to overcome contact angle hysteresis, the contact line of the IL droplet will be shifted and the IBFW patterns will be jeopardized. According to Liu et.al.⁶⁴, a negative bias voltage exceeds -1 V would induce significant contact angle decreases for [EMIM][NTf₂] on gold with contact line spreading forwards, which is also observed in our experiments. **A negative voltage smaller than -0.5 V, on the contrary, does not influence the contact angle obviously. Therefore, the deposition voltages adopted in current work does not influence the contact line position or the IBFW liquid film.**

Four cycles of cyclic voltammogram (CV) of 10 mM Ag⁺ in IL is shown in Fig. 4f. The reduction peak of Ag⁺ takes place at -0.29 V (vs. Ag) with the peak current decreases as the scan cycles increases. We believe that the micro litter droplet with limited analyte dissolved lead to such results. The Ag⁺ concentration decreases quickly after each cycle of CV scan, and the electron transfer is slower for the oxidation of Ag metal. In Fig. 4g, we test the CV curves of five scan cycles of IL droplet with 6 mM Ag⁺ and 5 mM TCNQ. Two reduction peaks can be distinguished, one at 0.095 V (vs. Ag), and the other at -0.21 V (vs. Ag). The first peak corresponds to the reduction of Ag⁺ to Ag⁰ (metal)⁴⁴, and the second peak is related to the formation of AgTCNQ complex (solid). Finally, we conceptually validate that IBFW technology is capable of transferring liquid film pattern into various solid materials and reveal the possibilities can be produced by combining IBFW with electrochemical procedures. The three-electrode experiment configuration, example of open-circuit-potential V-t curve and potentiostatic deposit I-t curve are shown in Supplementary Fig. 21.”

(Page 10-11, Main text)

“As summarized in Fig. 4h, the IBFW technology manifests several intriguing features that can be harnessed for a variety of application fields. The IBFW technique can fabricate patterned ILs film with 30 nm thickness, 100 nm spatial resolution and over hundreds of μm film length on insulating substrates (or coated with conducting metal films with 10¹ nm thickness). The surface-volume ratio endowed by the nanometer scale thickness can largely enhance the sensitivity of the IBFW film, and can be utilized in gas sensing circuit. **The good solubility and biocompatibility of ILs make them suitable for the dissolve of various analytes.**

The IBFW film also exhibits robustness against air environment exposure, gravity, and physical contact to droplet reservoir. Such features reveal that the IBFW film can act as stable flow channel for the analyte solutions injected to the reservoir, and can largely simplify the fabrication procedures of micro/nanofluidic chips. Last but not least, IBFW liquid film with wide potential window can be combined with electrochemical procedures and the patterned liquid film can be transformed into different solid particles. Such results demonstrate the IBFW as a versatile tool for both nanofluidics and liquid/solid materials printing.”

(Page 11, Main text)

Figure 4 (d) The schematics of a three-electrode electrochemistry experiment. A droplet of IL with IBFW liquid film pattern printed onto the working electrode is the solvent for analytes.

10 nm Au and 5 nm Ti deposited to 300 nm TOX SiO₂ serves as the working electrode, Pt probe and silver-plated probe stuck into the droplet are the counter and pseudo-reference electrodes respectively. **(e)** An example of the patterned solid particles deposited on Au electrode surface. The upper part shows Ag particles make up a 20 μm × 3 μm rectangular film with PKU letters pattern which are deposited to Au surface at -0.2 V (vs. Ag). The lower part shows blue AgTCNQ particles make up a 50 μm × 10 μm pattern which are deposited to Au surface at -0.1 V (vs. Ag). Both are potentiostatically deposited for 180 s. **(f)** Four cycles of cyclic voltammogram of 10 mM Ag⁺([EMIM][NTf₂]) solution. The reduction peak at -0.295 V (vs. Ag) can be seen, with peak current decreases with cycles. After four cycles, the reduction peak of Ag⁺ to Ag metal becomes less obvious. **(g)** Five cycles of cyclic voltammogram of 6 mM Ag⁺ and 5 mM TCNQ ([EMIM][NTf₂]) solution. The first reduction peak of Ag⁺ to Ag metal at 0.095 V (vs. Ag) can be seen. The +300 mV shift of Ag reduction peak at present of TCNQ is consistent with literature⁴⁴. The reduction peak current of Ag reduces quickly and become hard to distinguish as cycle increases. **(h)** A summary schematic to show the distinct features of IBFW technique, and the potential application fields that are suitable for IBFW.

(Page 27-28, Main text)

Supplementary Figure 21: The electrodeposition experiments of AgTCNQ.

(a) The three-electrode configuration adopted in current work. **(b)** The green line is the V-t curve of AgTCNQ open circuit potential measurement, which serves as a reference for the subsequent experiments. The purple line is the deposition current vs. time curve of AgTCNQ deposition at constant potential, -0.1 V (vs. Ag).

(Page S38, Supplementary)

Comment #2: *The surface-wetting driven printing process is limited to patterns that are*

continuously connected. For printing applications, most of the time disconnected patterns are needed. Are there ways to create such disconnected patterns (in the nano/micron scale)?

Response: We thank the reviewer for the important suggestions. By employing the more common feature of HFIB, which is to decompose and etch the sample, discontinuous liquid film patterns can also be printed. After the printing of a continuous liquid film, we use the NPVE software to scan a small region repeatedly to etch the unwanted part of the pattern to get a discontinuous pattern.

“Apart from continuous pattern, **discontinuous liquid pattern can be fabricated by the introduction of damaging mode of HFIB.** As shown in Supplementary Fig. 7, a PKU pattern is separated from the flow channel that connect to the droplet reservoir.”

(Page 4, Main text)

Supplementary Figure 7: Discontinuous pattern achieved by employing damaging mode of HFIB.

After the PKU letters pattern has been induced from a continuous liquid film channel, the conjunction parts between the film channel and the letters pattern are etched by HIM (by setting the scan dose density to 10^4 ions/nm², the liquid film is etched) .

(Page S23, Supplementary)

Comment: *If either or both of the above concerns can be successfully addressed, I suggest that this manuscript may be reconsidered. Otherwise, it may be more suitable to a more specialized journal.*

Response: In our resubmitted version of manuscript, we have attempted to address the

concerns about the application versatility of IBFW through two experiments that can broaden the liquid working media of IBFW and can transfer liquid pattern into solid materials. Discontinuous pattern can also be printed by combining the liquid inducing mode with damaging mode of the helium focused ion beam.

REVIEWERS' COMMENTS

Reviewer #1 (Remarks to the Author):

The authors have addressed my comments to satisfaction. I recommend publication.

Reviewer #2 (Remarks to the Author):

The authors have seriously considered all my previous comments and, in particular, improved the manuscript significantly based on them. In particular my main concern of lack of application potential was addressed with additional experiments and a new section in the manuscript.

I can recommend publication now.

Reviewer #3 (Remarks to the Author):

The authors have successfully addressed all of my previous concerns. I suggest publication as is.

Responses to Reviewer #1:

The authors have addressed my comments to satisfaction. I recommend publication.

Response: We thank the reviewer for the comments that have significantly improved our manuscript.

Responses to Reviewer #2:

The authors have seriously considered all my previous comments and, in particular, improved the manuscript significantly based on them. In particular my main concern of lack of application potential was addressed with additional experiments and a new section in the manuscript.

I can recommend publication now.

Response: We thank the reviewer for the comments that have significantly improved our manuscript.

Responses to Reviewer #3:

The authors have successfully addressed all of my previous concerns. I suggest publication as is.

Response: We thank the reviewer for the comments that have significantly improved our manuscript.